# The influence of the age-period-cohort effects on male suicide in Brazil from 1980 to 2019

**Weverton Thiago da Silva Rodrigues**[1], **Taynãna César Simões**[2], **Carinne Magnago**[3], **Eder Samuel Oliveira Dantas**[4], **Raphael Mendonça Guimarães**[5], **Jordana Cristina de Jesus**[1], **Sandra Michelle Bessa de Andrade Fernandes**[6], **Karina Cardoso Meira**[6]*

**1** Graduate Program in Demography at UFRN, Natal, Rio Grande do Norte, Brazil, **2** René Rachou Research Institute, Oswaldo Cruz Foundation of Minas Gerais, Belo Horizonte, Minas Gerais, Brazil, **3** School of Public Health, University of São Paulo, São Paulo, Brazil, **4** Onofre Lopes University Hospital of the Federal University of Rio Grande do Norte, Natal, Brazil, **5** Sergio Arouca National School of Public Health of the Oswaldo Cruz Foundation, Rio de Janeiro, Brazil, **6** School of Health, Federal University of Rio Grande do Norte, Natal, Rio Grande do Norte, Brazil

* karina.meira@ufrn.br

**Data Availability Statement:** The databases used in this work are available in the following doi:10.5281/zenodo.7547810 (https://zenodo.org/record/7547810#.Y8gJunbMLrd).

## Abstract

Suicide is a complex and multi-determined phenomenon. Higher rates are observed in men and are related to multiple risk factors, including mental disorders, financial crises, unemployment, and easy access to highly lethal means of perpetration, such as firearms. We studied the effects of age, period, and cohort (APC) on total and firearm-related suicides in men in Brazil and its major regions from 1980 to 2019. Death records were extracted from the Brazilian Ministry of Health's Mortality Information System. Estimable functions were used to estimate APC models, through the Epi library of the R statistical program, version 4.2.1. During the study period, Brazil had an average rate of 10.22 deaths per 100,000 men. Among regions, rates ranged from 8.62 (Northeast) to 16.93 (South). The same profile was observed in suicides by firearms. After estimating the APC models, we observed a temporal trend of increasing total suicides for Brazil and regions, except for the South region, where the trend was stationary. The trend was downward for firearm suicides for all locations. A positive gradient was observed in the mortality rate with advancing age for total suicides; and peak incidence between 20–29 years, with subsequent stabilization, for suicides perpetrated by firearms. There was a reduction in the risk of death for suicides perpetrated by firearms in relation to the reference period (1995–1999) for all locations, except in the North region, where the effect was not significant. The younger generations from the 1960s onwards had a higher risk of death from total suicide and a lower risk for those perpetrated by firearms in relation to the reference cohort (1950–1954). We observed a reduction in the mortality trend for suicides perpetrated by firearms, a reduction in the risk of death in the 2000s and for men born after 1960. Our results suggest reducing the risk of death from suicide by firearms in Brazil and regions. However, there is an upward trend in mortality from total suicides in the study period (1980–2019) and for younger cohorts.

**Funding:** The research was funded by the Coordination for the Improvement of Higher Education Personnel (CAPES) code 0001, which contributed with scholarships so that the author Weverton Thiago da Silva Rodrigues could attend his Master's Degree in the Graduate Program in Demography at the Federal University of Rio Grande do Norte. The funders had no role in study design, data collection and analysis, decision to publish, or preparation of the manuscript.

**Competing interests:** The authors have declared that no competing interests exist.

## Introduction

Suicide is a global public health problem, characterized as a complex, multidimensional and multidetermined social phenomenon that involves psychological, social, biological, environmental, political and cultural aspects. The latest estimates from the World Health Organization account for 703,000 deaths annually in the world. In summary, a death from this cause occurs every 42 seconds, representing an average global rate of 9.0 deaths per 100,000 inhabitants, of which 77% occur in low- and middle-income countries, such as Brazil [1].

Brazil is among the ten countries with the highest absolute numbers of suicides in the world. In 2019, there were more than 13,000 deaths from this cause, resulting in a national rate of 6.6 deaths per 100,000 people. Recent data indicate that the number of suicides increased in both sexes from 2010 to 2019, with higher rates in the states of Rio Grande do Sul, Santa Catarina and Piauí, respectively [2].

When consummated suicide is considered, this is a predominantly male problem. In Brazil, men have a 3.8 times greater risk of death by suicide than women. Considering the year 2019, the national suicide death rate in men was 10.7 per 100,000. Generally, this difference is explained by greater lethality of the means used by men and their greater intentionality to die [3].

The high numbers of death by suicide in men have also been associated with socioeconomic issues, as this population is more susceptible to the impacts of economic instabilities and greater access to firearms and other lethal objects [2, 3]. In a sociological study of suicide, Durkheim pointed to a type of suicide called anomic. For the author, when society is disturbed, either by economic and political crises or by wars and radical revolutions, it becomes incapable of exerting a moralization on individuals, provoking in them a strong feeling of disintegration and, consequently, an increase in suicide rates [4].

In addition, it is argued that compliance with masculinity standards has been associated with a worse mental health status and, mainly, with a lower search for mental health services among men, making them more susceptible to suicide. Thus, when considering that deaths by suicide in Brazil are predominantly male, multiple health authorities' efforts are needed to understand this phenomenon and propose control measures [3, 5]. Indeed, suicide attempts by men are generally rated as more severe regardless of the suicide methods used, suggesting gender differences in the intentionality associated with suicidal behavior [6].

In Brazil, some strategies were adopted in the field of health aimed at preventing suicide, especially after the 2000s. Since 2015, for example, the topic has been addressed more extensively in national campaigns, such as Setembro Amarelo (Yellow September)—a month dedicated to raising suicide awareness. In addition, in 2019, the National Policy for the Prevention of Self-Mutilation and Suicide was instituted [7, 8]. Another preventive measure was the Disarmament statute (2003), since it established stricter criteria for the sale of weapons and ammunition, in addition to regulating the registration of possession and carrying of weapons in the national territory [8, 9].

This public policy aimed to reduce homicides, suicides, and accidents with firearms, as it was believed that reducing the circulation of firearms in society would reduce the presence of firearms in homes, thus avoiding events of violence with firearms, more prevalent in households, such as suicide, violence against women and accidents involving children [8, 9]. Furthermore, it was believed that this public policy could reduce mortality from this health problem in all age groups. However, policies to control access to firearms have a cohort effect, as they impact different age groups unequally. Younger people may have difficulty accessing firearms, but middle-aged and elderly adults may already be carrying this instrument, and thus the effect of this preventive measure would be less in these age groups [10–12].

Thus, the present study aims to evaluate the temporal effects (age, period, and cohort) on overall and firearm suicides in Brazil and its major regions from 1980 to 2019. The analysis of the temporal trend of demographic and epidemiological indicators, such as the mortality rate, makes it possible to understand the behavior of diseases or health problems in different periods, enabling the assessment of public policies for prevention and control (period effect) and raising hypotheses about the effect of exposure (risk and protective factors) over time in different generations (birth cohort). In epidemiological research, it is common to come across this type of event ordered over time, resulting in three components that can contribute to the temporal evolution of the incidence and mortality of health problems: age, period, and cohort (APC): age, period, and cohort (APC) [13, 14].

In view of the above, the present study has the following research questions: Are there differences in the evolution of temporal effects (age, period, and cohort) in mortality from total suicide and suicide by firearms in men in Brazil and its five major regions, in the period from 1980 to 2019?

## Materials and methods

### Study design and data source

This is an ecological study of the temporal trend of mortality from total and firearm-related suicide in men in Brazil, and in its major regions, from 1980 to 2019. The country is divided into 26 states and a Federal District, which are grouped into five large regions (North, Northeast, Southeast, South and Midwest) with different demographic and socioeconomic characteristics and wide internal inequalities. The best human development indices are observed in the South, and the worst in the Northeast and North regions, the latter being characterized by its low population density and territorial extension that shelters a large part of the Amazon rainforest. The Southeast region, the most populous, stands out for the job market; while the Midwest, although it includes the capital of the country, has an economy focused on agriculture and livestock [15].

### Study variables

The study population consisted of men residing in Brazil and its major regions, aged from 14 years old, who died between 1980 and 2019 and had suicide as the underlying cause of death. The data source was the Mortality Information System of the Department of Informatics of the Unified Health System (SIM/DATASUS), of the Brazilian Ministry of Health, which provides death records for all Brazilian states and municipalities from 1979 to 2020, under the microdata format in the dbc extension [16].

The population data used for mortality estimates were also obtained from DATASUS, based on the demographic censuses of 1980, 1991, 2000 and 2010, and on population projections estimated in the intercensus years by the Brazilian Institute of Geography and Statistics [15].

For this study, the microdata were converted to the dbf extension using the Tabwin program, version 4.15 for Windows, provided by the Brazilian Ministry of Health. Subsequently, the records were exported and grouped for each region in the R software (version 4.2.1), according to age group and year, extracting only the records of interest. This extraction considered the ninth and tenth editions of the International Statistical Classification of Diseases and Related Health Problems (ICD-9 and ICD-10), according to the codifications presented in Table 1.

The Brazilian mortality information system presents limitations in the quality and coverage of death records, especially in locations with high socioeconomic vulnerability, [17–19]. Thus, to obtain more reliable estimates of mortality according to causes, the application of indirect correction techniques is recommended [13, 14]. Considering the lack of consensus in the

**Table 1. Death records from the Mortality Information System of the SUS Information Technology Department (SIM/DATASUS), according to ICD-9 and ICD-10.**

| Health problem | Coding in the ICD-9 | Coding in the ICD-10 |
|---|---|---|
| Suicide | E950 a E958 | X60 a X84 |
| Intoxication or self-intoxication unknown whether intentional or accidental | E980 a E982 | Y10 a Y19 |
| Sequelae of self-inflicted injury | E959 | Y87 |

literature regarding the best correction method for death records classified as suicide [20–24], in this study, the correction method used by the Brazilian Ministry of Health [25] was used and applied by Dantas *et al.* [21].

The process of rectifying the records was carried out by two independent researchers and checked by a third party, applying two steps for total suicides. In the first phase, the records of deaths originally classified as *suicide* were added to the records classified as intoxication or self-intoxication unknown whether intentional or accidental and sequelae of self-inflicted injury, according to age group, year, and region for the period from 1980 to 2019 [21, 25]. The next step consisted of correcting the coverage of deaths (underreporting) for males, according to Brazilian regions for the 1980s, 1990s, 2000s and 2010s, using the correction factors proposed by Lima *et al.* [26]. For each decade, these correction factors were multiplied by the number of deaths obtained in the previous step. For suicides perpetrated by firearms, correction was performed only for the underreporting of death records.

After correction of death records, mortality rates from total suicide and firearms were calculated, according to age group and geographic region, per 100,000 men, by age groups of five years. Truncated rates for open-range ages (80 years and over) were estimated per year. After obtaining the rates by age group and ages of the open interval, the five-year rates were standardized by the direct method, having the standard population proposed by the World Health Organization [27]. In addition, standardized total and firearm suicide rates were calculated. To descriptively assess the temporal trend of these health problems, mortality rates smoothed by three-year moving averages were calculated to correct for the random fluctuation present in annual mortality rates.

## Statistical analyses

In the analysis of temporal effects (APC), age groups and periods of the same size were chosen to avoid the occurrence of an identifiability problem due to artificial cyclic patterns when using periods and age groups of different sizes in the estimation [28–30]. Thus, $I = 15$ age groups from 10–14 years up to 80 years and older, $J = 8$ periods from 1980 to 2019, and $K = I + J − 1 = 22$ cohorts from 1900 to 2005 were analyzed.

The APC effects were estimated based on regression models, with Poisson distribution for the number of deaths observed in each age group i and period j ($\theta_{ij}$), the effects being additively related to the logarithm of the expected mortality rate ($E(r_{ij})$), according to Holford's proposal [28].

$$\ln\left(E\left(r_{ij}\right)\right) = \ln\left(\theta_{ij}/N_{ij}\right) = \mu + \alpha_i + \beta_j + \gamma_k$$

where $E[r_{ij}]$ represents the expected mortality rate at age group (*i*) and period (*j*); $\theta_{ij}$ is the number of deaths at age group (*i*) and period (*j*); $N_{ij}$ denotes the population at risk of death at age group (*i*) and period (*j*); μ represents the average rate; and $\alpha_i$ corresponds to the effect of age group (*i*), $\beta_j$ the effect of period (*j*), and $\gamma_k$ the effect of cohort *k* [28].

The main limitation of the estimation of APC effects is the problem of non-identifiable parameters in the complete model, due to the exact linear relationship between the temporal effects (i = j – k). There are several methodological proposals that propose to correct this limitation [14, 27–30]. In this study, the APC effects were estimated by the estimable functions, as proposed by Holford [29] and implemented in the models fitted by the Epi library (https://CRAN.R-project.org/package=Epi) from *sofware* R (https://www.R-project.org/) [31].

The estimable functions are limited to the analysis of linear combinations and the effects of curvatures of temporal terms. The curvatures are estimable functions of the parameters and remain constant regardless of the parameterization used. The linear trend of the effects is divided into two components: the linear effect of age and the drift effect (linear effect of period and cohort) [14, 29]. Longitudinal age trend is the sum of age and period slope (αL + βL), being αL and βL linear trends of age and period, respectively. The second drift term represents the linear trend of the logarithm of age-specific (mortality) rates and is equal to the sum of the period and cohort slopes. (βL + γL), being βL and γL period and cohort linear trends, in that order [14, 15, 27, 31].

Twelve APC analyzes (scenarios) were performed, resulting from the analysis for suicides without discrimination of the means of perpetration (total suicide) and suicides by firearm for Brazil and each of its geographic regions. In each scenario, the adjusted APC submodels were compared in a nested way via *deviance* statistics and the likelihood ratio test, at a 5% significance level, as proposed by Holford [28, 29].

The *deviance* analysis of the APC models determined by the Epi library estimates six nested equations:

(1): f(a) age;

(2): f(a) + δc age-drift;

(3): f(a) + h(c) age-cohort;

(4): f(a) + g(p) + h(c) age-period-cohort;

(5): f(a) + g(p) age-period; and

(6): f(a) + δp age-drift.

Where a, p and c represent the effects of age, period, and cohort; f, h and g represent smooth functions of the parameters; and δ is a linear effect [14, 30, 31].

The comparison of the submodels presented in the results section occurred in a nested way by age, age-*drift* (*drift*-cohort model), age-cohort, age-period-cohort, age-period, age-*drift* (*drift*-period model). This comparison was carried out by the likelihood ratio test between two submodels, namely: (1) it compares the age and age-*drift* models: if significant, it represents a nonlinear effect of age; (2) it compares age-*drift* and age-cohort models: if significant, it represents a nonlinear cohort effect; (3) it compares the age-cohort and age-period-cohort models, if significant, it represents a non-linear period effect, in the presence of a cohort; (4) it compares the age-period-cohort and age-period models: if significant, it represents a nonlinear cohort effect, in the presence of a period; (5) compares the age-period and age-*drift* models: if significant, it represents a non-linear period effect [14, 15, 27, 32, 33].

Based on the best-fit model, estimated age-specific mortality rates and relative risks (RR) were extracted for each period and cohort, as a function of the respective reference categories (period 1995–1999 and cohort of birth 1950–1954). Interval estimates were obtained at the 95% confidence level.

In our analyses, the period from 1995 to 1999 was the reference period because it was prior to the implementation of public policies that may have influenced the temporal trend of suicides and suicide attempts, such as the National Mental Health Policy, Disarmament Statute, income redistribution policies [7–9]. The reference cohort was from 1950–1954, as central cohorts tend to be more stable and complete than the first and last cohorts [14, 31], and previous studies carried out in Rio de Janeiro and in the Brazilian regions showed a lower risk of death in cohorts from the 1950s [32, 33].

After adjusting the APC models, the temporal trend of the mortality coefficients for general suicide and firearms suicides was analyzed by the linear age-drift trends and their respective 95% CI. The age-drift trend is stationary when the 95% confidence interval contains the value 1, descending when the age-drift and its 95% CI is less than 1, and ascending when the trend and the CI are greater than 1 [28–30].

## Ethical considerations

The data used in this study were freely accessed from the SIM/DATASUS [13]. There are no identified individuals in this system. Therefore, this study was not submitted to a Research Ethics Committee.

## Results

In the period under study, Brazil had 231,821 deaths from suicide in men (9.05 deaths per 100,000 men) and 36,895 deaths from suicides perpetrated by firearms (2.55 deaths per 100,000 men). A greater percentage variation of increase in male total suicide was observed after the correction stages in the North (+43.05%) and Northeast (+27.92%) regions and the lower in the South (+6.95%) and Southeast (+11.44%) regions (Table 2).

**Table 2. Standardized mortality rates per 100,000 men Brazil and its major regions, in five-year periods, in the period 1980–2019.**

| Locality | 1980 to 84 | 1985 to 89 | Suicide 1990 to 94 | 1995 to 99 | 2000 to 04 | 2005 to 09 | 2010 to 14 | 2015 to 2019 | SAR[a] |
|---|---|---|---|---|---|---|---|---|---|
| Brazil | | | | | | | | | |
| UMR | 7.35 | 7.20 | 7.96 | 9.30 | 8.86 | 9.34 | 9.56 | 10.78 | 9.05 |
| TC | 8.98 | 8.76 | 9.14 | 10.68 | 10.11 | 10.03 | 11.05 | 12.32 | 10.22 |
| North | | | | | | | | | |
| UMR | 4.06 | 4.62 | 5.19 | 5.91 | 6.33 | 7.30 | 8.05 | 9.70 | 7.01 |
| TC | 6.92 | 7.74 | 7.17 | 9.34 | 8.68 | 9.89 | 11.20 | 13.07 | 10.03 |
| Northeast | | | | | | | | | |
| UMR | 3.23 | 3.21 | 4.36 | 6.22 | 6.35 | 8.12 | 8.38 | 9.79 | 6.74 |
| TC | 4.67 | 4.58 | 5.58 | 7.03 | 8.11 | 10.37 | 10.63 | 12.29 | 8.62 |
| Southeast | | | | | | | | | |
| UMR | 7.44 | 7.03 | 7.61 | 8.37 | 7.54 | 7.81 | 8.47 | 8.47 | 8.02 |
| TC | 8.66 | 8.22 | 8.59 | 9.38 | 8.35 | 8.60 | 9.39 | 9.92 | 8.94 |
| South | | | | | | | | | |
| UMR | 14.99 | 14.91 | 15.38 | 18.60 | 16.51 | 15.75 | 14.46 | 16.58 | 15.83 |
| TC | 17.08 | 16.52 | 16.58 | 19.91 | 17.45 | 16.50 | 15.20 | 17.49 | 16.93 |
| Midwest | | | | | | | | | |
| UMR | 6.51 | 7.55 | 9.22 | 13.52 | 12.05 | 11.26 | 11.16 | 12.54 | 10.81 |
| TC | 9.97 | 11.16 | 10.78 | 13.90 | 13.77 | 12.81 | 12.90 | 14.44 | 12.83 |

[a]SAR = Standardized Average Rate;

[b]UMR = Uncorrected mortality Rate;

[c]TC (total correction) = Mortality rates corrected for information quality and under-enumeration of deaths.

A national trend towards an increase in mortality rates due to total suicides was observed, especially in the older age groups and from the 2000s onwards, with a suggestion of an increase in mortality rates from the 1950 cohort for the younger age groups (Fig 1). Similarly, there was a progressive increase in the average mortality rates from total suicides with advancing age in all geographic regions and a reduction in the coefficients towards the younger generations. An increase in rates was also observed from 1995–1999 in all locations, except for the South region, where there was a reduction and subsequent increase (Fig 2).

Mortality rates smoothed by three-year moving averages suggest an increase in suicide mortality rates for Brazil from 1999 onwards, similar to that observed in the Northeast region. In the North and Southeast regions, the increase is noticeable from 2004 onwards, and in the Midwest from the beginning of the 2000s, with constant growth from 2007 onwards. In the South region, we observe rates increase followed by reductions; however, from 2010, there was a progressive increase in rates (S1 Fig).

Similarly, there was a progressive increase in the average mortality rates from total suicides with advancing age in all geographic regions and a reduction in the coefficients towards the younger generations. An increase in rates was also observed from 1995–1999 in all locations, except for the South region, where there was a reduction and subsequent increase (Fig 2).

Regarding suicides by firearms, a reduction in mortality rates was identified in almost all age groups from the late 1990s and the 1950 cohort for younger age groups in Brazil and regions (Figs 3 and 4). Firearm suicide rates smoothed by three-year moving averages show a reduction in mortality rates in Brazil and all its major regions in the late 1990s (1997 to 1999) or early 2000s (2002 to 2003) (S2 Fig). There was a progressive increase in the average mortality rates of suicides by firearms with advancing age in the Northeast, South, and Midwest regions and a reduction in the North and Southeast regions. In the latter, the lowest value observed was in the 80 and over age group (Fig 3).

Tables 3 and 4 show the *deviance* analysis of age-adjusted models for period and cohort. The best-fit model for total suicides and for suicides perpetrated by firearms was the one that contained the three temporal effects (APC model).

After adjusting for the APC models, the linear trend of age together with period (age-drift) showed upward for mortality rates for total suicides, for all regions except for the South, whose trend was stationary. Inversely, a downward trend was observed for all regions for suicides perpetrated by firearms (Table 5).

Differences were found between the average rates according to age group adjusted for the effect of the period and cohort for total suicides and by firearm. For total suicides, there was a progressive increase in mortality rates with advancing age, except in the Southeast region, which showed a reduction in the age groups from 35 to 44 years, with a subsequent increase and stabilization at around 11.0 deaths per 100,000 men (Figs 5 and 6). In suicides by firearms, in all locations, a mortality peak was identified in the age group of 25–29 years, with a subsequent reduction and stabilization for Brazil and the Northeast and Midwest regions. In the North region, there was an increase in mortality in the 25–29 age group, with a subsequent reduction. In the South, after the reduction in the age groups from 30–34 to 40–44 years, there was a progressive increase in the following age groups (Figs 5 and 6).

The period effect adjusted for the effect of age and cohort, showed a reduction in the risk of death in relation to the reference period (1995–1999) for all five-year periods in Brazil. The exceptions were for the five-year periods 1990–1994 (RR = 1.012; 95%CI1.009–1.015) and 2015–2019 (RR = 1.016; 95%CI1.004–1.028) for total suicides (Fig 5).

In the North region, there was an increased risk of death in the periods from 1980 to 1994 and 2010 to 2019 for total suicides. For firearm suicides, the risk was not statistically significant in any five-year period. A similar profile was identified in the Northeast region, where the risk

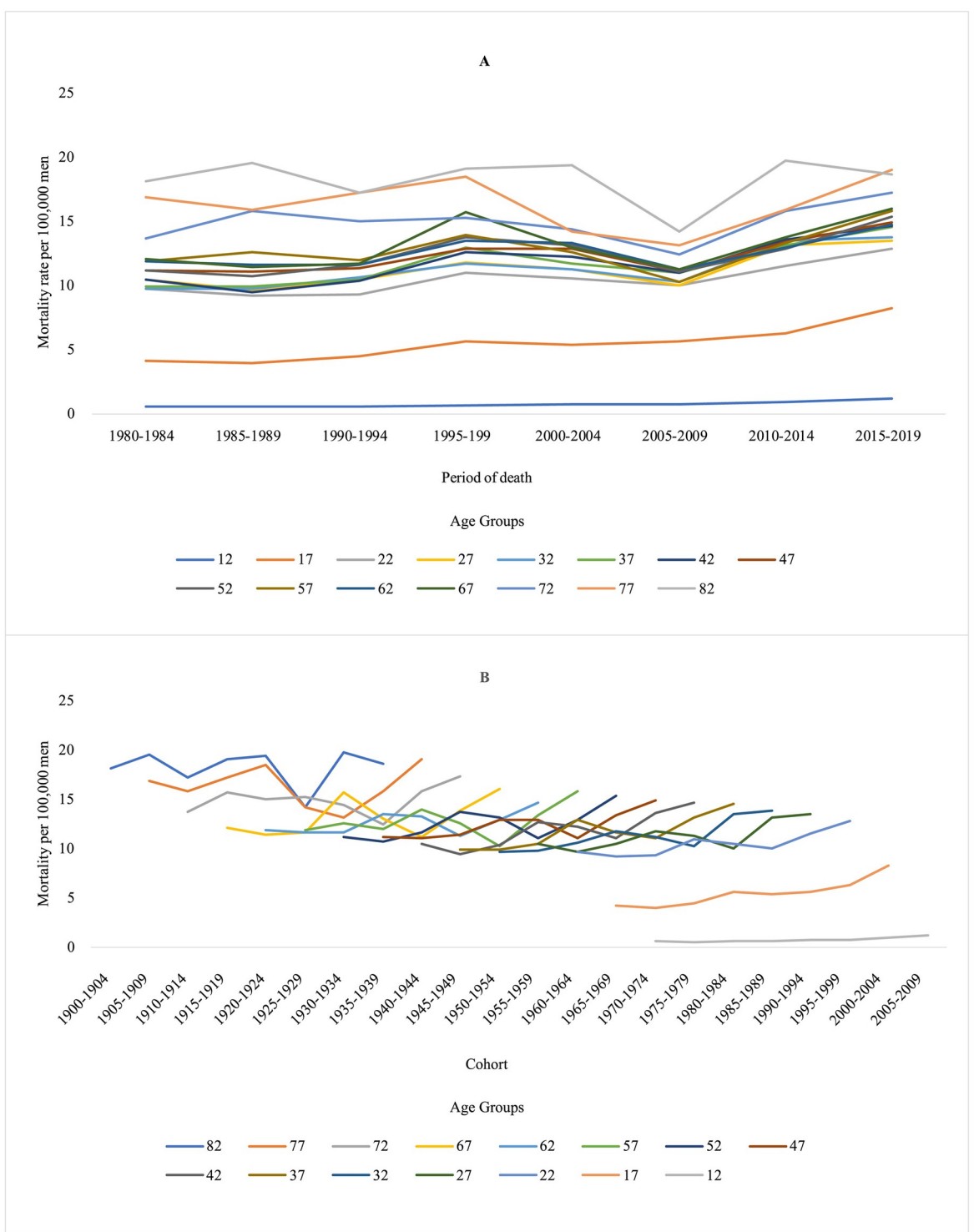

**Fig 1. Total Suicide mortality rates per 100,000 men in Brazil, according to age group, period, and cohort, from 1980 to 2019.**

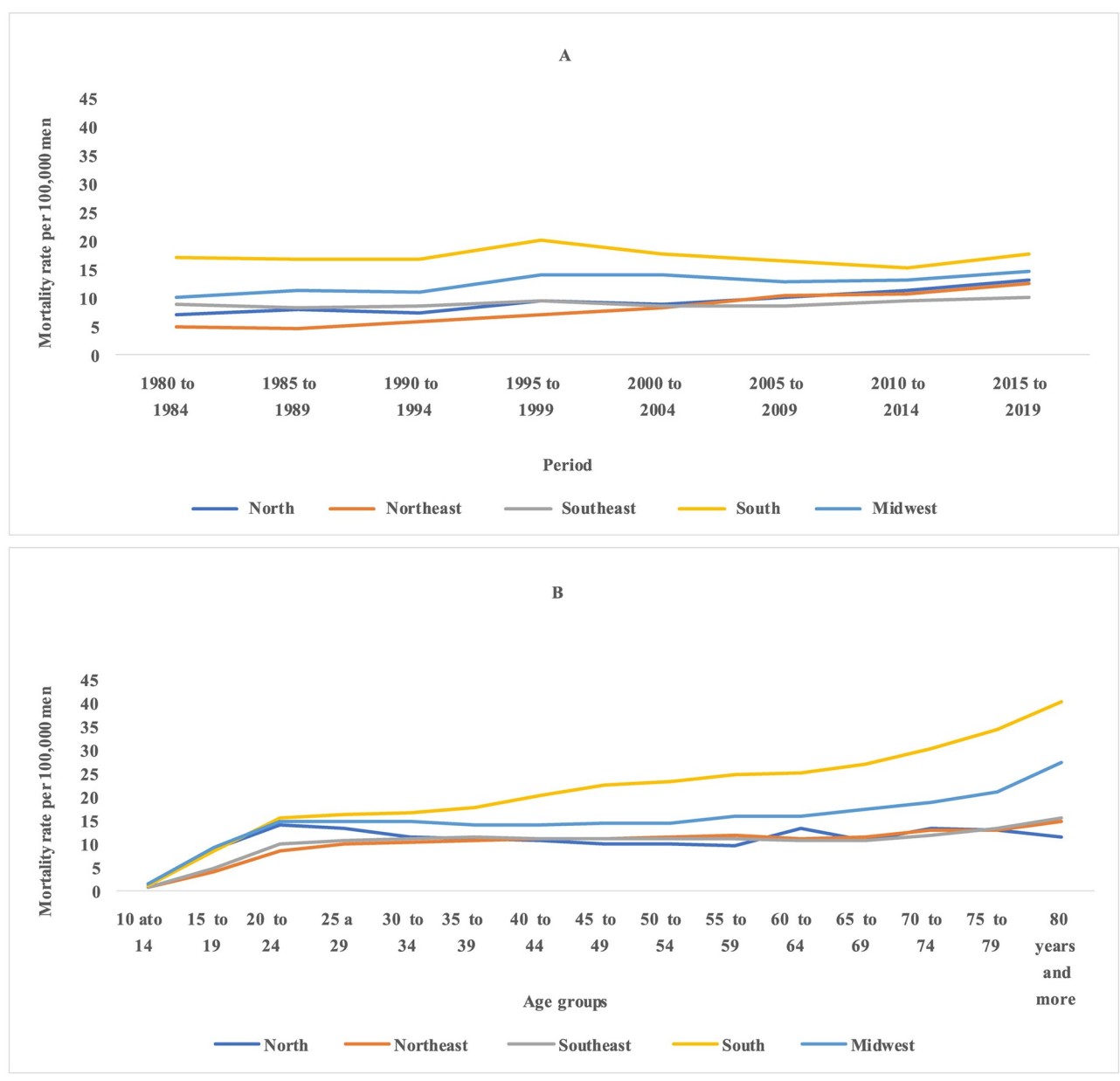

**Fig 2. Total Suicide mortality rates per 100,000 men in in major Brazilian regions according to age, period, and cohort, from 1980 to 2019.**

increased in the 1980s and in the period from 2000 to 2009 for the total of suicides. On the other hand, there was a reduction in the risk of death by suicide by firearm for all periods, when compared to the reference period. In the Southeast region, there was an increase in risk in the periods 1990–1994 and 2015–2019, when total suicides were analyzed, and reduction in the risk of death from suicide by firearm. In the South and Midwest, there was a protective effect (RR<1) for total and firearm-related suicides for all periods in relation to the reference period (1995–1999), except for the period 2000–2004 (Fig 7).

The cohort effect adjusted for the effect of age and period, showed a reduction in the risk of death in relation to the reference period, showed a protective effect (RR<1) for cohorts prior to 1950–1954 (reference cohort) and an increase in later generations for total suicides for

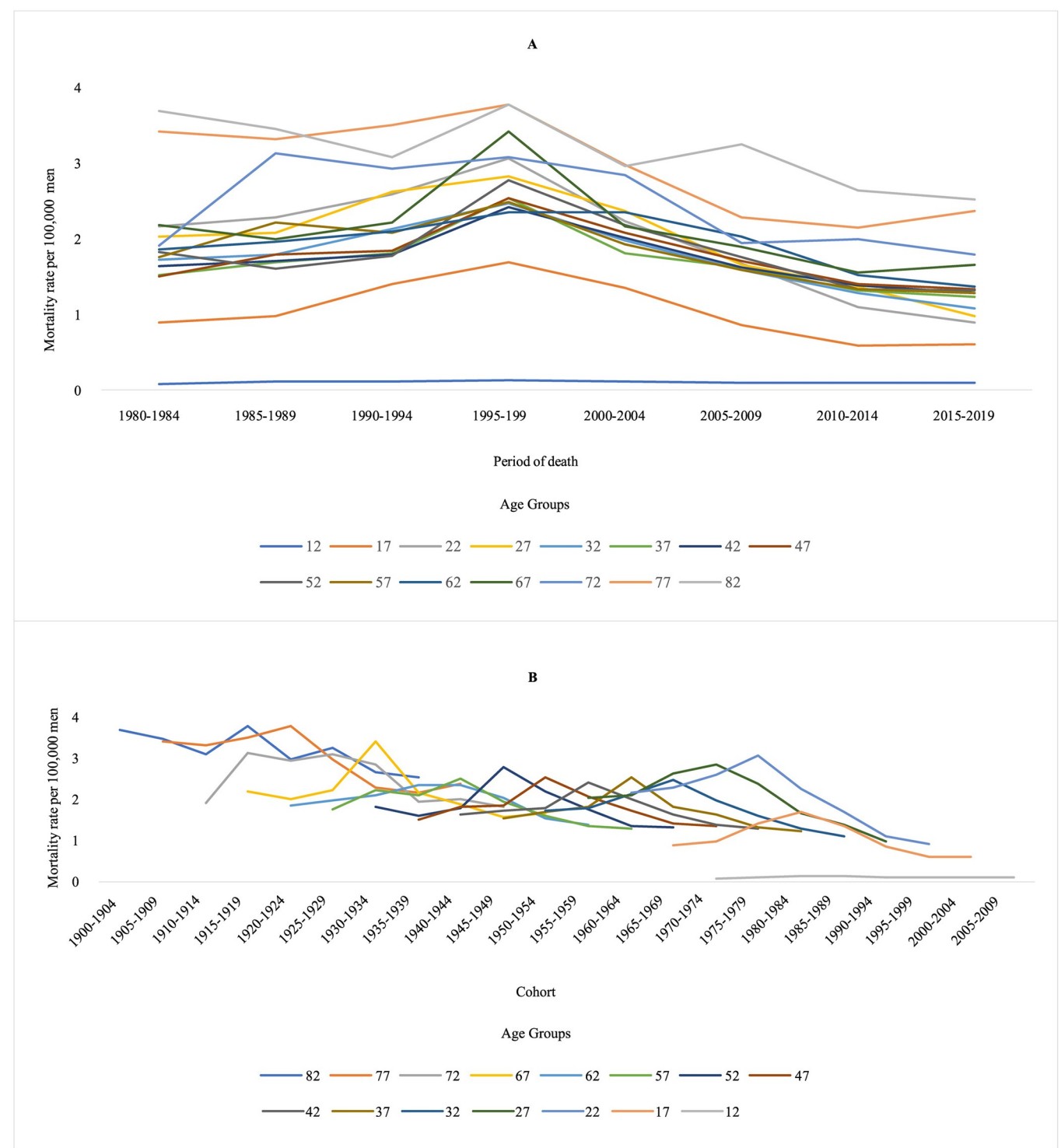

**Fig 3. Suicide by firearms mortality rates per 100,000 men in Brazil, according to age group, period, and cohort, from 1980 to 2019.**

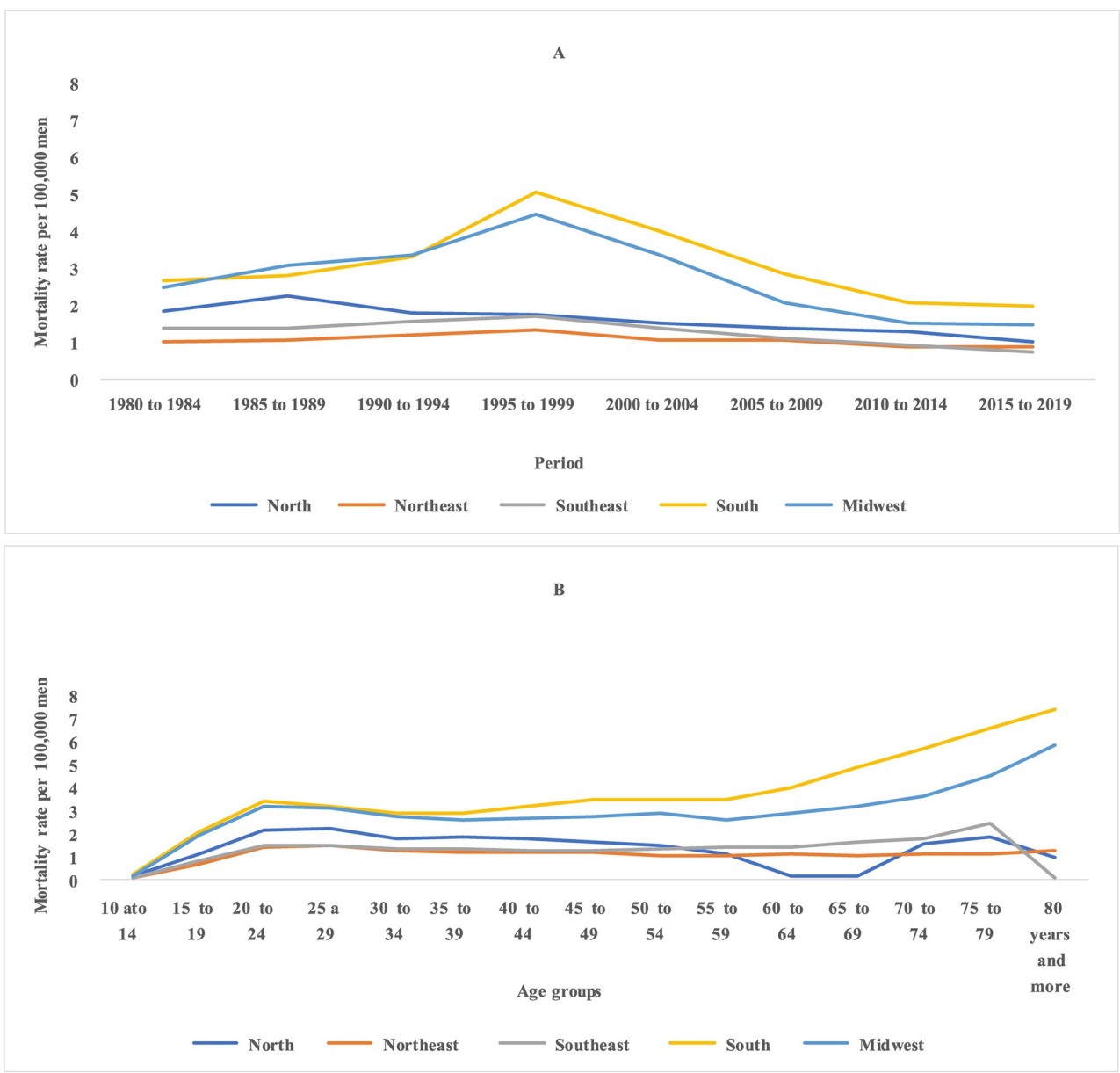

**Fig 4. Suicide by firearms mortality rates per 100,000 men in in major Brazilian regions according to age, period, and cohort, from 1980 to 2019.**

Brazil and all regions. However, in the South and Southeast regions there was also RR>1 for generations prior to the reference. Regarding suicides by firearms, an increase in risk was identified in cohorts prior to 1950–1954 and a reduction in all subsequent generations. In the Northeast region, the reduction was significant only for men born from 1975 to 2009 (Figs 5 and 8).

## Discussion

Suicide is a multicausal phenomenon that varies according to psychological, social, biological, and cultural issues in a given society. However, globally, men are at higher risk of death by suicide than women. Authors argue that this reality has a strong correlation with situations in

**Table 3. Deviance an p-value analysis in sequential construction of age, period, and cohort models for total suicides in men, according to Brazil its major regions, from 1980 to 2019.**

| Locality | | | |
|---|---|---|---|
| | Brazil | | |
| Models | Df[a] | Deviance Residual | p (>\|Chi\|) |
| Age | 115 | 15370.75 | |
| Age-drift | 114 | 12799.09 | <0.00001 |
| Age-Cohort | 111 | 12351.44 | <0.00001 |
| Age-Period- Cohort | 108 | 11690.76 | <0.00001 |
| Age-Period | 111 | 12030.81 | <0.00001 |
| Age-drift | 114 | 12799.09 | <0.00001 |
| | North | | |
| Modelos | Df[a] | Deviance Residual | p (>\|Chi\|) |
| Age | 115 | 1748.73 | |
| Age-drift | 114 | 899.48 | <0.00001 |
| Age-Cohort | 111 | 774.40 | <0.00001 |
| Age-Period- Cohort | 108 | 764.44 | <0.00001 |
| Age-Period | 111 | 853.68 | <0.00001 |
| Age-drift | 114 | 899.48 | <0.00001 |
| | Northeast | | |
| Age | Df[a] | Deviance Residual | p (>\|Chi\|) |
| Age-drift | 115 | 8282.75 | |
| Age-Cohort | 114 | 2459.19 | <0.00001 |
| Age-Period- Cohort | 111 | 2418.88 | <0.00001 |
| Age-Period | 108 | 2274.89 | <0.00001 |
| Age-drift | 111 | 2313.61 | <0.00001 |
| Age-drift | 114 | 2459.19 | <0.00001 |
| | Southeast | | |
| Age | Df[a] | Deviance Residual | p (>\|Chi\|) |
| Age-drift | 115 | 5189.92 | |
| Age-Cohort | 114 | 4945.41 | <0.00001 |
| Age-Period- Cohort | 111 | 4623.20 | <0.00001 |
| Age-Period | 108 | 4508.44 | <0.00001 |
| Age-drift | 111 | 4829.88 | <0.00001 |
| Age-drift | 114 | 4945.41 | <0.00001 |
| | South | | |
| Age | Df[a] | Deviance Residual | p (>\|Chi\|) |
| Age-drift | 115 | 3879.20 | |
| Age-Cohort | 114 | 3875.95 | <0.00001 |
| Age-Period- Cohort | 111 | 3764.64 | <0.00001 |
| Age-Period | 108 | 3530.84 | <0.00001 |
| Age-drift | 111 | 3662.38 | <0.00001 |
| Age-drift | 114 | 3875.95 | <0.00001 |
| | Midwest | | |
| Age | Df[a] | Deviance Residual | p (>\|Chi\|) |
| Age-drift | 115 | 1313.37 | |
| Age-Cohort | 114 | 1146.13 | <0.00001 |
| Age-Period- Cohort | 111 | 1127.40 | <0.00001 |
| Age-Period | 108 | 1067.17 | <0.00001 |

(*Continued*)

**Table 3.** (Continued)

| Locality | | | |
|---|---|---|---|
| Age-drift | 111 | 1095.81 | <0.00001 |
| Age-drift | 114 | 1146.13 | <0.00001 |

<sup>a</sup>Degrees of freedom.

which men are unable to perform hegemonic masculinity, such as the loss of economic power during crises of capitalism, in retirement, diseases that generate disabilities and dependence The white, cisgender, heterosexual, sexually active, productive, and prosperous man is considered as a representative of hegemonic masculinity. This pattern of masculinity ends up oppressing both women and other men (black, gay, migrant, physically disabled or poor) [34–45].

In the period 2000–2019, the global suicide rate decreased by 36%, with drops ranging from 17% to 49% between different regions of the planet. The Americas region was the only one to show an upward rate, reaching 17% in the same period. Compared with developed countries in this and other regions, Brazilian suicide rates are considered intermediate. However, in absolute numbers, we occupy the eighth place [2, 46, 47], expressing the importance of this phenomenon for public health and Brazilian society.

In our study, the average Brazilian male suicide rate was 10.2 per 100,000 men, from 1980 to 2019, with an upward trend in mortality in all geographic regions, except for the South, where the trend remained stable. It was also this region that presented the highest mortality rates in the different five-year periods analyzed, while the Northeast and Southeast presented the lowest. Previous investigations had already shown this profile for the main means of perpetration: hanging, self-intoxication and firearms [2, 34, 46, 47].

It is believed that the magnitude and transcendence of this problem is greater, not being fully captured by information systems due to underreporting and the high proportion of suicides recorded as events of undetermined cause. That is, death is recorded, but not always its cause, especially in higher socioeconomic classes, where stigma is usually greater and there are practical issues, such as life insurance. Thus, in response to requests from family members, many coroners register death by suicide as an event of undetermined cause [2, 21–24]. In this study, we sought to minimize these limitations by correcting the records and underreporting of deaths, as recommended by the Brazilian Ministry of Health [25].

After the correction process, the less developed Brazilian regions (North and Northeast) showed the highest percentages of increase in the number of deaths, while the South and Southeast regions, the most developed, showed the lowest percentages. The correlation between the quality of death records and regional socioeconomic development confirms the need to use correction techniques when assessing the temporal trend over a long period and in different locations [14, 15, 19, 48].

In Brazil and its major regions, throughout the analyzed period, the most used means of committing suicide was hanging, followed by firearms and self-intoxication, as identified in previous national and international investigations [2, 34, 40, 41, 46, 47]. The choice of means of self-extermination is related to the victim's gender and the availability of the instrument to be used. Men tend to choose more lethal methods such as hanging and the use of a firearm, while women more often opt for self-intoxication [2, 34, 49]. When performing hegemonic masculinity, men intend to demonstrate strength and virility, so they opt for methods whose probability of survival is lower. Failure in their self-extermination process, for them, would be read as weakness or a typically female act "to draw attention" [3, 50].

**Table 4. Deviance an p-value analysis in sequential construction of age, period, and cohort models for suicides by firearms in men, according to Brazil its major regions, from 1980 to 2019.**

| Locality | | | |
|---|---|---|---|
| | Brazil | | |
| Age | Df[a] | Deviance Residual | p (>\|Chi\|) |
| Age-drift | 115 | 5610.93 | |
| Age-Cohort | 114 | 4444.92 | <0.00001 |
| Age-Period- Cohort | 111 | 3863.52 | <0.00001 |
| Age-Period | 108 | 2768.71 | <0.00001 |
| Age-drift | 111 | 3014.51 | <0.00001 |
| Age-drift | 114 | 4444.92 | <0.00001 |
| | North | | |
| Age | Df[a] | Deviance Residual | p (>\|Chi\|) |
| Age-drift | 115 | 381.718 | |
| Age-Cohort | 114 | 237.976 | <0.00001 |
| Age-Period- Cohort | 111 | 229.197 | <0.00001 |
| Age-Period | 108 | 223.507 | <0.00001 |
| Age-drift | 111 | 229.542 | <0.00001 |
| Age-drift | 114 | 237.976 | <0.00001 |
| | Northeast | | |
| Age | Df[a] | Deviance Residual | p (>\|Chi\|) |
| Age-drift | 115 | 557.75 | |
| Age-Cohort | 114 | 493.79 | <0.00001 |
| Age-Period- Cohort | 111 | 444.34 | <0.00001 |
| Age-Period | 108 | 388.47 | <0.00001 |
| Age-drift | 111 | 410.41 | <0.00001 |
| Age-drift | 114 | 493.79 | <0.00001 |
| | Southeast | | |
| Age | Df[a] | Deviance Residual | p (>\|Chi\|) |
| Age-drift | 115 | 1989.28 | |
| Age-Cohort | 114 | 1402.87 | <0.00001 |
| Age-Period- Cohort | 111 | 1182.73 | <0.00001 |
| Age-Period | 108 | 912.06 | <0.00001 |
| Age-drift | 111 | 1031.6 | <0.00001 |
| Age-drift | 114 | 1402.87 | <0.00001 |
| | South | | |
| Age | Df[a] | Deviance Residual | p (>\|Chi\|) |
| Age-drift | 115 | 2143.33 | |
| Age-Cohort | 114 | 1928.99 | <0.00001 |
| Age-Period- Cohort | 111 | 1725.97 | <0.00001 |
| Age-Period | 108 | 995.56 | <0.00001 |
| Age-drift | 111 | 1073.80 | <0.00001 |
| Age-drift | 114 | 1928.99 | <0.00001 |
| | Midwest | | |
| Age | Df[a] | Deviance Residual | p (>\|Chi\|) |
| Age-drift | 115 | 1151.16 | |
| Age-Cohort | 114 | 809.65 | <0.00001 |
| Age-Period- Cohort | 111 | 686.71 | <0.00001 |
| Age-Period | 108 | 385.03 | <0.00001 |

(*Continued*)

**Table 4.** (Continued)

| Locality | | | |
|---|---|---|---|
| Age-drift | 111 | 431.06 | <0.00001 |
| Age-drift | 114 | 809.65 | <0.00001 |

[a]Degrees of freedom.

It is suggested that the temporal evolution of suicide is intrinsically related to changes in the age structure, form of social, cultural, and economic organization of a given society [14, 15]. That said, and to assist in raising hypotheses about the mechanisms involved with the suicide [14, 15, 41], we estimated the effect of age, period, and birth cohort on total suicide and suicide by firearms, in Brazil and its major regions in the period from 1980 to 2019.

There was a positive gradient in the national rates of total suicide with advancing age, for suicide by firearm there was a peak incidence between 20–29 years, with subsequent stabilization. The higher magnitude of firearm suicide rates in this age group was confirmed after adjusting for period and cohort effects. Similar findings had already been documented for Brazil in the period from 1980 to 2015 [31]. In other countries, however, a higher risk of self-harm by this means of perpetration has been observed in elderly men [43, 51, 52].

Authors conjecture that the greatest risk of suicide in the elderly, regardless of the means of perpetration, is related to social isolation caused by retirement, loss of family members and spouses, loss of social status, and reduced purchasing power due to low pension values. [38–44]. Diseases that generate disabilities, such as chronic, degenerative, and neurological diseases, the diagnosis of HIV/AIDS, among other pathologies that generate intense pain and anguish, are also important risk factors for suicidal behavior. Individuals with these pathologies are three times more likely than the general population to present such behavior [1, 53].

**Table 5.** Linear age-drift trend for Brazil and its geographic regions after fitting the APC model by estimable functions.

| Locality | | | |
|---|---|---|---|
| Brazil | Age -drift | CI-95% | Trend |
| Total suicide | 1.008 | 1.007–1.009 | Upward |
| Suicide by firearm | 0.984 | 0.983–0.985 | Downward |
| North | Age-drift | CI-95% | Trend |
| Total suicide | 1.02 | 1.019–1.021 | Upward |
| Suicide by firearm | 0.979 | 0.976–0.983 | Downward |
| Northeast | Age-drift | CI-95% | Trend |
| Total suicide | 1.03 | 1.029–1.032 | Upward |
| Suicide by firearm | 0.991 | 0.989–0.993 | Downward |
| Southeast | Age-drift | CI-95% | Trend |
| Total suicide | 1.004 | 1.004–1.005 | Upward |
| Suicide by firearm | 0.981 | 0.980–0.983 | Downward |
| South | Age-drift | CI-95% | Trend |
| Total suicide | 0.999 | 0.998–1.000 | Stationary |
| Suicide by firearm | 0.998 | 0.986–0.989 | Downward |
| Midwest | Age -drift | CI-95% | Trend |
| Total suicide | 1.007 | 1.006–1.009 | Upward |
| Suicide by firearm | 0.975 | 0.973–0.978 | Downward |

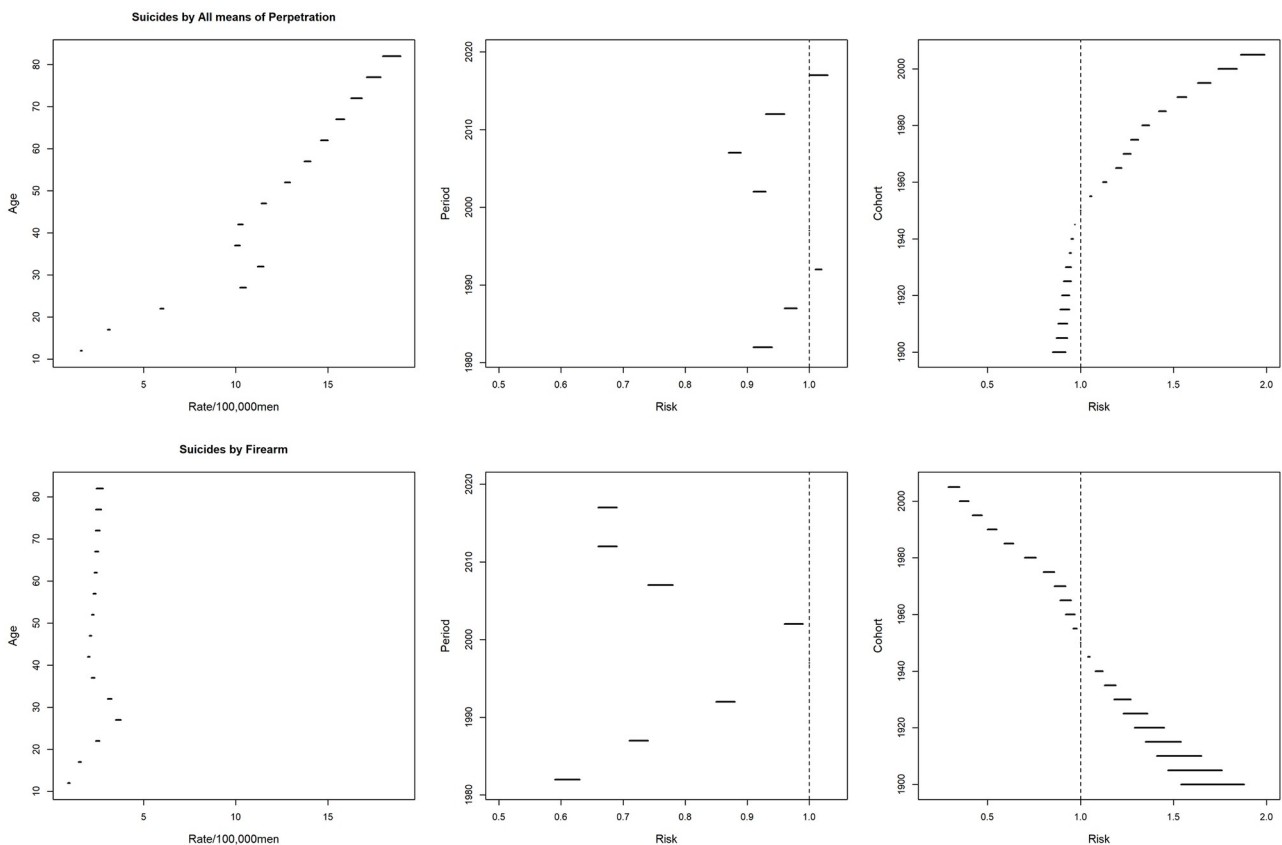

**Fig 5. Age, period, and cohort effects estimated by estimable functions for total and firearm-related suicides, in Brazil, from 1980 to 2019.**

Although suicide rates can be between six and eight times higher among the elderly compared to the young, the increase in rates in the latter group seems to be a worldwide trend [1, 54]. In Brazil, we expected a more significant reduction in firearm suicide rates among men aged 20 to 24 compared to other age groups. The Disarmament Statute raised the age for purchasing weapons from 21 to 25 years. However, our findings show higher rates in the 20–24 to 25–29 age groups, suggesting that restricting legal access to firearms has not been sufficient to promote a reduction in suicide in younger age groups. We conjecture that this subpopulation has greater ease of access and purchase in the illegal market, including online—a hypothesis that needs to be investigated in future studies. Thus, policies are needed to control arms trafficking in the country, intersectoral policies to prevent adolescents and young people from getting involved in organized crime, associated with public policies that increase youth participation in sports and cultural activities, especially in high-income territories—socioeconomic vulnerability [10, 11, 55, 56].

Regarding the analysis of period effects (social, political, economic, cultural events, diagnostic and therapeutic advances of a given period and location that impact all age groups) and taking as a reference the five-year period 1995–1999, we verified a distinct profile between total suicides and firearm suicides. There was a reduction in the risk of death (RR<1) from suicide by firearm for all periods and locations, except for the North region, where the risk was not significant for any five-year period. As for the risk of total suicide it was reduced in the South and Midwest regions in all periods, and in the Southeast region, in the first decade of the 2000s. On the other hand, it was increased for Brazil (2015–2019) and for the North (1980–1994 and

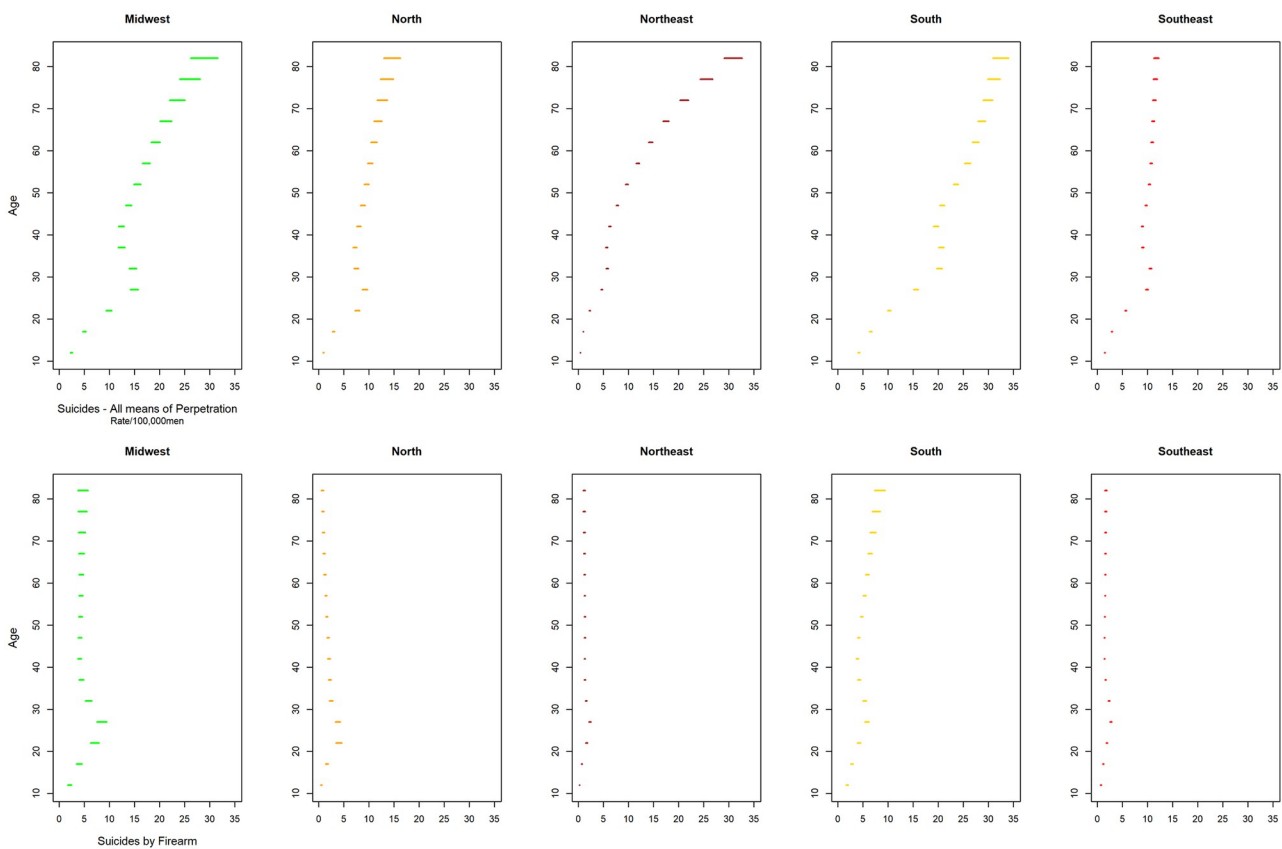

**Fig 6. Mortality rates by age group, per 100,000 men, adjusted for period and cohort effect, for total suicide by firearms, from 1980 to 2019, Brazil and its major regions.**

2005–2015) and Southeast (1980–1994 and 2015–2019) regions. Furthermore, it produced an upward age-drift trend for the entire period under study for total suicide, excluding the South region, whose trend remained stable. It is noteworthy that the present descriptive analysis showed a reduction in mortality due to firearm suicide in Brazil and regions prior to implementing the Disarmament Statute, with the most significant decrease in the last five years (2015–2019). These results suggest that the reduction in suicides by firearms is not only correlated with the Disarmament Statute, but other factors may also have contributed to this reality; future studies should be carried out to understand this reality better.

Authors advocate that measures that reduce the availability of lethal means of perpetrating suicide can reduce suicide rates in communities [1, 6, 48, 49]. However, studies have shown that in the absence of methods, such as firearms, individuals can migrate to alternative methods [10, 11, 12], which may explain the increase in mortality rates from total suicides in the 2000s and younger cohorts, even with a reduction in the risk of death from suicide by firearms, observed in the present study.

Young men are considered more impulsive and use lethal means to commit suicide. Barriers to accessing firearms, this group of people usually resorts to hanging to commit suicide [10, 11, 12], an easily accessible method of perpetration, which only has effective prevention measures in institutionalized people. In this direction, a study carried out in Brazil from 1980 to 2019 showed an increase in mortality rates due to suicide by hanging of 10.12% comparing the 1990s with the 2000s and an increase of 51.97% in the coefficients of the period from 2000

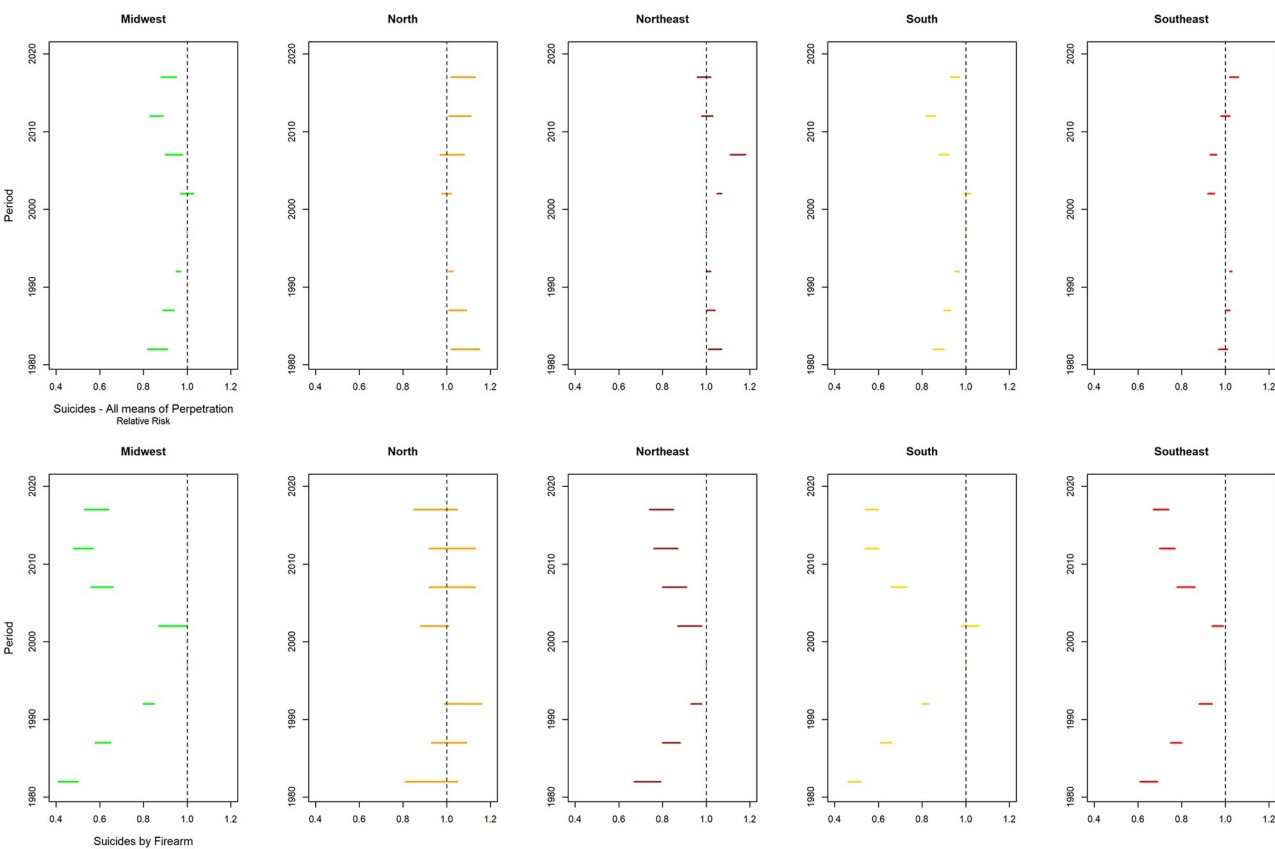

**Fig 7. Risk of death (relative risk) by period, adjusted for age and cohort effect, for total suicide and suicide by firearms, from 1980 to 2019, Brazil and major regions.**

to 2009 compared to the period from 2010 to 2019. In the same periods, there was a reduction in firearm suicide rates, respectively, 16.92% and 29.63%. Similar results were observed in all Brazilian regions [57]. However, this study only performed a descriptive analysis, so further studies are needed to corroborate the method substitution hypothesis.

It is essential to ratify that only an isolated preventive measure, such as control of access and circulation of firearms, is not enough to reduce overall suicide rates, and other measures are necessary. Studies in suicidology state that to effectively reduce suicides in countries organized national public policies must be promoted and focus on improving social and economic conditions and access to health for the population [1, 48, 49].

During the last four decades (1980–2019), Brazil has experienced an accelerated urbanization process, an increase in the proportion of youth and adults in the population and changes in family composition. In the same period, it went through important and long economic cycles of recession and economic stagnation [58–60]. The longest economic crisis lasted for 25 quarters (1981–1987), during which time there was a change in the official currency and economic plan, and high inflation. In the following decade, there were periods of recession between 1996 and 1999. After a growth phase in the first decade of the 2000s, a new cycle of recession took place because of the 2008 global economic crisis, with a reduction in gross domestic product in 2014, which lasted for eleven quarters [61–63]. The current economic recession has been accompanied by fiscal adjustment measures, with reduced spending on health and education and withdrawal of labor and social security rights. [64–67].

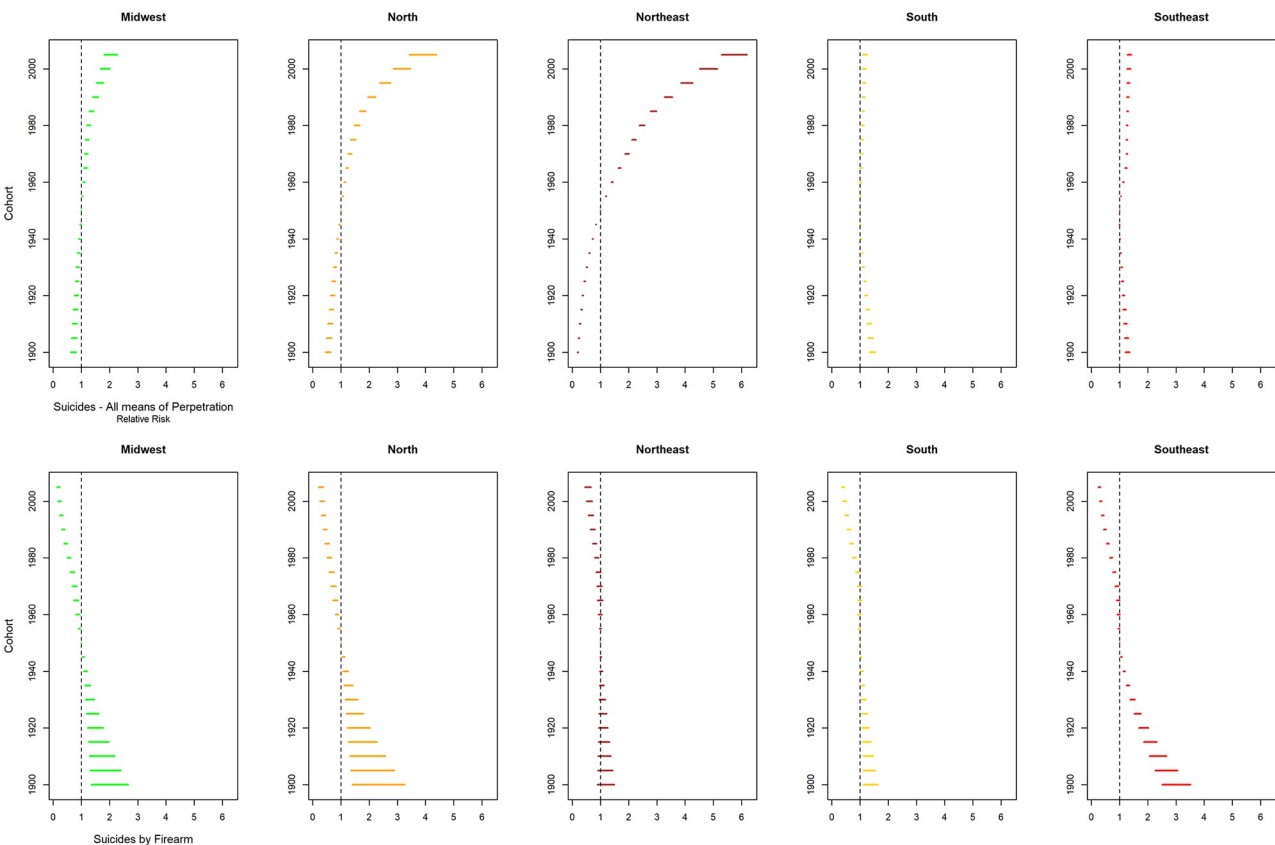

**Fig 8. Risk of death (relative risk) by cohort, adjusted for age and period effect, for total suicide and suicide by firearms, from 1980 to 2019, Brazil and major regions.**

Economic crises cause imbalance and disturbances in the collective order of societies, reducing social integration and control [4] and increasing social isolation, cognitive, behavioral, and biological despair. For these reasons, they are associated with an increase in the prevalence of mental disorders, such as depression, bipolar affective disorder, and anxiety disorders. These, in turn, are strongly associated with an increased risk of diseases of despair, such as suicidal ideation, illicit drug use, alcohol and other drug abuse, and deaths from despair (suicide, overdose, and liver cirrhosis) [60, 68–70].

In this perspective, studies have shown an increased risk of suicide in the period of economic crises in different countries and continents. The crisis in Southeast Asian countries at the end of the 1990s promoted an increase in suicides in 1998 and 1999 and in the first five years of the 2000s in Japan and South Korea. In Hong Kong, the increase occurred after the health crisis of the SARS epidemic (2003); in the United States it took place from the economic instability of 2008; in Russia it took place during the period of economic regime changes in the 1990s [40–42].

In theory, as they constitute a period effect, economic crises have an impact on all age groups. However, studies have shown greater effects in men in the fourth and fifth decades of life, who essentially make up the working class. Considering that this group assumes economic responsibilities for the younger and older generations, it suffers with greater intensity the consequences of unemployment, loss of income and the reduction of the living conditions of their families [40–44]. In this direction, research carried out in Australia showed a

correlation between the significant increase in the mortality rate from suicide in the 35–44 age group with the unemployment and underemployment rate in the 1990s. During this period, the country experienced a reduction in the supply of full-time jobs, an increase in part-time jobs and an increase in social inequalities [71]. Likewise, a population-based cohort survey in Finland showed an association between job instability and suicide rates. Individuals without a steady job had twice the risk of suicide compared to those with steady employment [72].

The increased risk of suicide in men from the late baby boomer generation (1956–1964) has been widely documented in the literature in several countries around the world, usually occurs in the fourth and fifth decade of life, being associated with economic crises, unemployment, reduced purchasing power, and labor and social security reforms with loss of rights [35, 36, 46, 51–53, 69, 71, 74]. In countries of the Global South that did not experience a Welfare State, the increased risk of suicide among those born into Generation X (1965 to 1979), late Y (1973–1980) and Millenials (1981–1991) may be related to the process of cohort discontinuity, which consists of the absolute increase in the number of members of certain generations due to the accelerated process of demographic transition in these countries [72–75].

When composed of many members, generational groups have their standard of living reduced as they experience greater competition for quality education, a more competitive job market with lower wages, and low pensions when they retire. Thus, throughout their life cycle, they will be at greater risk for social isolation as they fail to meet their expectations of life and success. This impact has been greater in developing countries with minimal state [72–77].

Generations that experience greater socioeconomic deprivation, with fewer ties of integration and social control, will have a higher risk of suicide compared to other cohorts that experience better living conditions, confirming the Durkheimian theory [4, 40, 41].

In our study, the increased risk of suicide in Millennials (1981 to 1995) and in Generation Z (1995 to 2009) is due to the increase in mortality in the age groups 15 to 34 years in the last two decades. Young people born between 1990 and 2000 experienced the cohort effect of the emergence of the internet and social media, which contributed to the contraction of face-to-face interactions and the expansion of the feeling of loneliness [78, 79]. Consequently, the prevalence of sadness, anxiety, feeling of not belonging, and suicidal behavior increased [67, 76, 77]. Furthermore, they are more exposed to early parental loss due to parental separation and, consequently, to reduced income and abusive use of alcohol and other drugs; and more subject to the *Werther* effect [39, 44, 78, 79].

The quality of information about suicide can be considered a limitation of this study since it can interfere with the temporal trend of suicides. However, more reliable estimates were obtained after the correction process was applied, reducing the impact of this counterpoint. Another limitation to be considered is the problem of identifiability of the complete model because of the correlation between the three effects (age, period, and cohort). Therefore, in this study, we used the estimable functions to generate probabilistic models, as recommended in the literature [14, 15, 28–31].

Furthermore, it is important to highlight that the effects of age, period and cohort have different impacts according to race and ethnicity. However, Brazil only has mortality data by race/skin color from 1996 onwards, and official bodies do not provide population estimates according to race/color and age group for the intercensus years, making it impossible to estimate APC models broken down by this variable.

The main contribution of this study resides in the fact that it evaluated mortality from overall suicide and suicide by firearms under the focus on temporal factors (age, period, and cohort), for Brazil and its major regions, after the correction of death records. Moreover, it showed that reducing firearm suicide rates from the end of the 1990s and in the cohorts from

the 1960s onwards was not enough to reduce mortality from total suicides in the same period and younger cohorts.

These results point to the possibility of replacing suicide methods. This hypothesis should be studied in future studies, using interrupted time series or quasi-experimental studies using analysis or differences-in-differences, including socioeconomic and demographic variables. Furthermore, studies are needed to assess the impact of the COVID-19 pandemic on suicide mortality in men according to temporal factors (age, period, and cohort) and methods used. An increase in suicide rates is expected due to social isolation, fear of the unknown, fear of contaminating oneself or family members, socioeconomic impacts, such as unemployment, food insecurity, and lack of medical assistance to manage psychosocial problems [70, 80, 81].

The diseases of despair are correlated with unemployment, low income, food insecurity, and difficulty accessing Public Health, Education, and Social Assistance Policies. If intersectoral public policies are not implemented to remedy these problems, suicide rates in Brazil may maintain an upward temporal trend [60, 68–70].

## Conclusion

The present study showed an increase in overall suicide in Brazil and its major regions and a reduction in these coefficients for suicides perpetrated by firearms, especially from the 2000s onwards. We also found a reduction in the risk of death in younger cohorts for self-extermination by firearms and an increase in total suicides.

Reducing access to lethal means of suicide is an important initiative to reduce suicides. However, this protective measure alone is not enough to reduce suicide rates, as methods may be replaced, especially in individuals with psychiatric illnesses who are more impulsive [10–12]. To prevent suicides in Brazil are needed to public policies increase access to employment and income, psychosocial care, and educational programs in schools to teach children and adolescents to improve their coping strategies to deal with stressful situations [10, 11]. In addition to training health professionals, family members, and educators to identify individuals at risk for diseases of despair so that they can be referred for treatment promptly.

## Informed consent statement

The data used in this research are secondary data of universal access in which there is no identification of the subjects, and thus it was not submitted to the Research 561 Ethics Committee according to Resolution 196/96 of the National Health Council.

## Supporting information

**S1 Fig. Total suicide rates in Brazilian men, smoothed by three-year moving means, according to major Brazilian regions, from 1980 to 2019.**
(TIF)

**S2 Fig. Firearm suicide rates among Brazilian men, smoothed by three-year mean averages, according to major Brazilian regions, from 1980 to 2019.**
(TIF)

**S3 Fig. Suicide by hanging rates among Brazilian men, smoothed by three-year mean averages, according to major Brazilian regions, from 1980 to 2019.**
(TIF)

## Author Contributions

**Conceptualization:** Weverton Thiago da Silva Rodrigues, Taynãna César Simões, Karina Cardoso Meira.

**Data curation:** Weverton Thiago da Silva Rodrigues, Karina Cardoso Meira.

**Formal analysis:** Taynãna César Simões, Karina Cardoso Meira.

**Investigation:** Weverton Thiago da Silva Rodrigues, Karina Cardoso Meira.

**Methodology:** Weverton Thiago da Silva Rodrigues, Taynãna César Simões, Raphael Mendonça Guimarães, Karina Cardoso Meira.

**Project administration:** Karina Cardoso Meira.

**Resources:** Karina Cardoso Meira.

**Software:** Weverton Thiago da Silva Rodrigues, Karina Cardoso Meira.

**Supervision:** Weverton Thiago da Silva Rodrigues, Taynãna César Simões, Carinne Magnago, Eder Samuel Oliveira Dantas, Raphael Mendonça Guimarães, Jordana Cristina de Jesus, Sandra Michelle Bessa de Andrade Fernandes, Karina Cardoso Meira.

**Validation:** Weverton Thiago da Silva Rodrigues, Taynãna César Simões, Carinne Magnago, Eder Samuel Oliveira Dantas, Raphael Mendonça Guimarães, Jordana Cristina de Jesus, Sandra Michelle Bessa de Andrade Fernandes, Karina Cardoso Meira.

**Visualization:** Weverton Thiago da Silva Rodrigues, Taynãna César Simões, Carinne Magnago, Eder Samuel Oliveira Dantas, Raphael Mendonça Guimarães, Jordana Cristina de Jesus, Sandra Michelle Bessa de Andrade Fernandes, Karina Cardoso Meira.

**Writing – original draft:** Weverton Thiago da Silva Rodrigues, Taynãna César Simões, Carinne Magnago, Eder Samuel Oliveira Dantas, Raphael Mendonça Guimarães, Jordana Cristina de Jesus, Sandra Michelle Bessa de Andrade Fernandes, Karina Cardoso Meira.

**Writing – review & editing:** Weverton Thiago da Silva Rodrigues, Taynãna César Simões, Carinne Magnago, Eder Samuel Oliveira Dantas, Raphael Mendonça Guimarães, Jordana Cristina de Jesus, Sandra Michelle Bessa de Andrade Fernandes, Karina Cardoso Meira.

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
