## [Decision Letter · Decision Letter 0]

6 Nov 2022

PONE-D-22-26688The influence of the age-period-cohort effects on male suicide in Brazil from 1980 to 2019PLOS ONE

Dear Karina Cardoso Meira,

Thank you for submitting your manuscript to PLOS ONE!

After careful consideration, we invite you to submit a revised version with substantial changes of the manuscript - that addresses the points raised during the review process – specially in the Discussion’ section.

We look forward to receiving your revised manuscript.

Kind regards,

Fernanda Penido Matozinhos, Ph.D

Academic Editor

PLOS ONE

Journal Requirements:

" No. The funders had no role in study design, data collection and analysis, decision to publish, or preparation of the manuscript."

Reviewers' comments:

Reviewer's Responses to Questions

**Comments to the Author**

1. Is the manuscript technically sound, and do the data support the conclusions?

Reviewer #1: No

Reviewer #2: Partly

2. Has the statistical analysis been performed appropriately and rigorously? 

Reviewer #1: No

Reviewer #2: No

3. Have the authors made all data underlying the findings in their manuscript fully available?

Reviewer #1: No

Reviewer #2: No

4. Is the manuscript presented in an intelligible fashion and written in standard English?

Reviewer #1: Yes

Reviewer #2: Yes

5. Review Comments to the Author

Reviewer #1: As a descriptive report of male suicide in Brazil through four decades, this manuscript adds little to knowledge on the theme. The Introduction discussed the topic in general terms, from Durkheim to disarmament policies, from trends in time series analysis to assessing age, period, and cohort effects. Methods repeated the explanation of APC models already offered in other papers by the same authors. Results collated too much information to be discussed in one only article. The Discussion is challenging to follow because the period is too large to comprise one single process (the disarmament hypothesis), and the spatial data analysis considered too vast geographical regions. To conclude, I noticed that the authors lost control of numbering the references.

As an academic exercise of data gathering and analysis, this manuscript adds little, if anything at all, to the professional field. Exercising is profitable for those who exercise; it is not a matter to be communicated to the international reader of health research.

Reviewer #2: A lot of work has gone into the APC calculations and the interpretation and discussion of the results. It is well written.

A significant issue with this study is that it is not a quasi-experimental design study such as time series analysis or differences-in-differences analysis. No independent variable such as social and economic factors have been incorporated in the statistical analysis. No sensitivity tests were performed using the dates the Disarmament Act was implemented. Hence no real casual arguments or even possible associations can really be made from this study. It is purely descriptive. The authors attempt to explain why some of the changes were observed but other factors than those the authors propose could in responsible.

Introduction:

1. Please provide some explanation of how the Disarmament Statute would reduce suicides. From reading around the legislation, it only increases the age of purchase from 21 to 25. Older men can still obtain firearms. Why would it reduce suicide in that cohort?

Results:

1. Line 250: could the increase in suicide rates be due to better record acquisition? Does your method correct for this?

2. Please correct the labels on the tables to English. Some are translated and some are not.

Discussion

1. Please provide a discussion about why the total suicide rate seemed to have increased and yet firearms suicide seems to have decreased in terms of substitution effect or displacement effect, that is if firearms are harder to obtain people switch to other methods leaving overall rates unchanged. That is while firearms restrictions may lower the rate of firearms suicide it doesn’t lower suicide rates overall. (Langmann 2021, Langmann 2020, Gilmour et al 2018)

2. It appears that in many countries the proportions or methods of suicide have switched to hanging. Could it be that there is an overall preference change to hanging that has nothing to do with the Disarmament Statute? (De Leo et al 2003)

3. It appears to me that suicide by firearms rate decreases start around 1998 (Figure 3) which is 6 years before the start of the Disarmament Statute (2004). Presumably it would also take some time to register firearms etc. When were the majority of firearms registered? It seems that the reduction in suicide by firearm occurred quite a period of time before the implementation of the legislation suggesting another cause may be responsible.

4. If the Disarmament Statute raise the age of acquisition from 21 to 25 should a larger reduction in suicide by firearm appear in that group than other groups as this group was prohibited from acquiring firearms?

5. Line 489: this study cannot demonstrate causation and therefore any claims that the Disarmament Statue was able to reduce suicide by firearms cannot be made. Line 522 last sentence, this claim can also not be made.

Figure 1 etc.

1. Please provide a legend to determine what the different colors represent etc.

6. PLOS authors have the option to publish the peer review history of their article (what does this mean?). If published, this will include your full peer review and any attached files.

Reviewer #1: No

Reviewer #2: No

---

## [Author Response · Author response to Decision Letter 0]

18 Jan 2023

Dear Editor,

We would like to take this opportunity to thank the anonymous

reviewers that have taken part in this review process for their comments regarding the

manuscript.

All comments were relevant and incorporating the suggested changes has improved the manuscript. The new revised version of the the manuscript has followed the recommendations of PlosOneObservation: The sentences in yellow have been removed or replaced by sentences in

green. Some sentences were added in blue.

PONE-D-22-26688

The influence of the age-period-cohort effects on male suicide in Brazil from 1980 to 2019

PLOS ONE

Dear Karina Cardoso Meira,

Thank you for submitting your manuscript to PLOS ONE!

After careful consideration, we invite you to submit a revised version with substantial changes of the manuscript - that addresses the points raised during the review process – specially in the Discussion’ section.

Journal Requirements:

Informed consent statement. 

The data used in this research are secondary data of universal access in which there is no identification of the subjects, and thus it was not submitted to the Research 561 Ethics Committee according to Resolution 196/96 of the National Health Council.

" No. The funders had no role in study design, data collection and analysis, decision to publish, or preparation of the manuscript."

No. The funders had no role in study design, data collection and analysis, decision to publish, or preparation of the manuscript. The research was funded by the Coordination for the Improvement of Higher Education Personnel (CAPES) code 0001, which contributed with scholarships so that the author Weverton Thiago da Silva Rodrigues could attend his Master's Degree in the Graduate Program in Demography at the Federal University of Rio Grande do Norte.

4. We note that you have stated that you will provide repository information for your data at acceptance. Should your manuscript be accepted for publication, we will hold it until you provide the relevant accession numbers or DOIs necessary to access your data. If you wish to make changes to your Data Availability statement, please describe these changes in your cover letter and we will update your Data Availability statement to reflect the information you provid

The databases used in this work are available in the following doi:10.5281/zenodo.7547810 (https://zenodo.org/record/7547810#.Y8gJunbMLrd)

Reviewers' comments:

Reviewer's Responses to Questions

Comments to the Author

1. Is the manuscript technically sound, and do the data support the conclusions?

Reviewer #1: No

Reviewer #2: Partly

2. Has the statistical analysis been performed appropriately and rigorously?

Reviewer #1: No

Reviewer #2: No

3. Have the authors made all data underlying the findings in their manuscript fully available?

Reviewer #1: No

Reviewer #2: No

4. Is the manuscript presented in an intelligible fashion and written in standard English?

Reviewer #1: Yes

Reviewer #2: Yes

5. Review Comments to the Author

Reviewer #1: As a descriptive report of male suicide in Brazil through four decades, this manuscript adds little to knowledge on the theme. The Introduction discussed the topic in general terms, from Durkheim to disarmament policies, from trends in time series analysis to assessing age, period, and cohort effects. Methods repeated the explanation of APC models already offered in other papers by the same authors. Results collated too much information to be discussed in one only article. The Discussion is challenging to follow because the period is too large to comprise one single process (the disarmament hypothesis), and the spatial data analysis considered too vast geographical regions. To conclude, I noticed that the authors lost control of numbering the references.

As an academic exercise of data gathering and analysis, this manuscript adds little, if anything at all, to the professional field. Exercising is profitable for those who exercise; it is not a matter to be communicated to the international reader of health research.

Response

We appreciate the reviewer's note. However, we ask permission to disagree with some aspects. First, we mention the context of Durkheim's study, as it is a pioneering study on suicide. It served as a reference for decades in the study of contextual effects and is classically the study used to describe distortions between empirical evidence of population aggregates and individual effects – the so-called ecological fallacy (Perace, 2000; Diez Roux, 2004; Harris et al, 2015). We consider its citation suitable to demonstrate that we know the limitation of ecological analyses, typically found in time series studies. It is essential to recognize that the burden of disease related to external causes in Latin America is so significant that we have adopted the term "triple burden of disease" for the region, following the example of India (except that India uses the expression to highlight the deaths in the pregnancy-puerperal cycle and infant mortality) (Lester, 1994; Ladusingh et al, 2018). We also emphasize that the recent trend towards violent deaths in Brazil results from public health policies aimed at this outcome and with more vulnerable populations. Concerning these vulnerable populations, we address particularly men, whose mortality differential is around 10 times the magnitude presented by women, a long term and well-known relationship (van de Venne et al, 2020). This relationship is a huge challenge in Latin America, and it avoid countries to converge epidemiological transition into a common point (Alvarez et al, 2020). These policies only exist thanks to diagnoses carried out over the last few decades in Brazil. For this reason, we believe that the analysis we carry out is not just an exercise in applying a method but rather a vital diagnosis to guide policies with a targeted target audience efficiently. 

The explanation is eventually repetitive regarding the method because it is the technique usually employed. The model's assumptions do not change depending on the outcome, and we try to be strict about fulfilling the assumptions required for modeling. Since the group that carried out the study has a long history in its application, it is reasonable to assume that the replication of the formalization of the model is similar to what we have already done in previous studies. Still, on the method, the study of a long period is necessary to make the method itself viable, which uses five-year intervals for the analysis. In addition, the option for comprehensive regions also considered two factors: the exclusion of the imprecision present in studies that work with small areas (there are hundreds of municipalities with zero counts of suicides, regardless of age). Furthermore, Brazilian regions reflect, to some extent, inequality at the subnational level. We consider that the manuscript presents a national diagnosis and can lead to more local analyses, including using data collected in loco.

We agree that the discussion is challenging. Not only because the problem at hand requires intersectional policies, but also because we propose to assess the temporal effects (APC) for total and firearm suicides after correcting death records for quality and underreporting of death records. However, we believe that we have achieved this goal by assuming the limitations of our study..Finally, we disagree about the lack of interest to the international reader. On the contrary, suicide and other mental health outcomes have gained significant notoriety over the past 15 years. This phenomenon started with the global economic crisis in 2008 (Alvarez-Galvez et al, 2021) and has increased with the Covid-19 pandemic (Demirci et al, 2020; Constanza et al, 2021).

REFERENCES

1. Alvarez JA, Aburto JM, Canudas-Romo V. Latin American convergence and divergence towards the mortality profiles of developed countries. Popul Stud (Camb). 2020 Mar;74(1):75-92.

2. Alvarez-Galvez J, Suarez-Lledo V, Salvador-Carulla L, Almenara-Barrios J. Structural determinants of suicide during the global financial crisis in Spain: Integrating explanations to understand a complex public health problem. PLoS One. 2021 Mar 1;16(3):e0247759.

3. Costanza A, Amerio A, Aguglia A, Serafini G, Amore M, Macchiarulo E, Branca F, Merli R. From "The Interpersonal Theory of Suicide" to "The Interpersonal Trust": an unexpected and effective resource to mitigate economic crisis-related suicide risk in times of Covid-19? Acta Biomed. 2021 Oct 1;92(S6):e2021417. 

4. Demirci Ş, Konca M, Yetim B, İlgün G. Effect of economic crisis on suicide cases: An ARDL bounds testing approach. Int J Soc Psychiatry. 2020 Feb;66(1):34-40.

5. Diez Roux AV. The study of group-level factors in epidemiology: rethinking variables, study designs, and analytical approaches. Epidemiol Rev. 2004;26:104-11. 

6. Harris AH, Humphreys K, Finney JW. State-level relationships cannot tell us anything about individuals. Am J Public Health. 2015 Apr;105(4):e8.

7. Ladusingh L, Mohanty SK, Thangjam M. Triple burden of disease and out of pocket healthcare expenditure of women in India. PLoS One. 2018 May 10;13(5):e0196835. 

8. Lester D. Gender equality and the sex differential in suicide rates. Psychol Rep. 1994 Dec;75(3 Pt 1):1162. 

9. Pearce N. The ecological fallacy strikes back. J Epidemiol Community Health. 2000 May;54(5):326-7.

10. van de Venne J, Cerel J, Moore M, Maple M. Sex Differences in Mental Health Outcomes of Suicide Exposure. Arch Suicide Res. 2020 Apr-Jun;24(2):158-185. 

Reviewer #2: A lot of work has gone into the APC calculations and the interpretation and discussion of the results. It is well written.

Response

A significant issue with this study is that it is not a quasi-experimental design study such as time series analysis or differences-in-differences analysis. No independent variable such as social and economic factors have been incorporated in the statistical analysis. No sensitivity tests were performed using the dates the Disarmament Act was implemented. Hence no real casual arguments or even possible associations can really be made from this study. It is purely descriptive. The authors attempt to explain why some of the changes were observed but other factors than those the authors propose could in responsible.

Response

We performed the analysis with the APC model to verify the three effects, mainly the cohort effect. We were able to identify the effects separately, and we detected that there was a period effect. This is the diagnosis that this technique offers. We consider it necessary to deepen. However, this is an underlying analysis that deserves a more profound debate.

Furthermore, this analysis requires the application of a specific technique (e.g., interrupted time series). It would be an oversight on our part to make this a detail in this manuscript rather than dedicating a complete analysis exclusively to this purpose. Therefore, at the end of the discussion, we indicate the possibility of unfolding without committing ourselves to carry out this in our study.

We believe that our research questions may have hinted that we would carry out a quasi-experimental study, and thus, in the present revision of the manuscript, so that the objectives, research questions and title following the same direction would make a change in the research question, which is described below.

“Thus, the present study aims to evaluate the temporal effects (age, period, and cohort) on total and firearm suicides in Brazil and its major regions from 1980 to 2019.

In view of the above, the present study has the following research questions: Are there differences in the evolution of temporal effects (age, period, and cohort) in mortality from total suicide and suicide by firearms in men in Brazil and its five major regions, in the period from 1980 to 2019?”

We believe that these modifications reflect what we are looking for and we can conclude from the method used (APC). Our primary independent variable is chronological time and its decomposition into age, cohort, and period effects. 

The reviewer's correct observation of the issue does not invalidate our study. On the contrary, it reinforces the need for contextual effects, the need for which could only be identified from our research and not a priori. In this sense, we also reinforce that we did not intend to identify causality effects. For this reason, we have made modifications to the discussion and conclusion suggested by the reviewer, so that there is no doubt about the impossibility of assessing causality with the APC method.

Introduction:

1. Please provide some explanation of how the Disarmament Statute would reduce suicides. From reading around the legislation, it only increases the age of purchase from 21 to 25. Older men can still obtain firearms. Why would it reduce suicide in that cohort?

We appreciate the valuable contribution of the reviewer. Added the paragraph below in the introduction.

Added in text

Another preventive measure was the Disarmament statute (2003), since it established stricter criteria for the sale of weapons and ammunition, in addition to regulating the registration of possession and carrying of weapons in the national territory [8,9]. This public policy aimed to reduce homicides, suicides, and accidents with firearms, as it was believed that reducing the circulation of firearms in society would reduce the presence of firearms in homes, thus avoiding events of violence with firearms, more prevalent in households, such as suicide, violence against women and accidents involving children [8,9]. Furthermore, it was believed that this public policy could reduce mortality from this health problem in all age groups. However, policies to control access to firearms have a cohort effect, as they impact different age groups unequally. Younger people may have difficulty accessing firearms, but middle-aged and elderly adults may already be carrying this instrument, and thus the effect of this preventive measure would be less in these age groups [10,11,12]

Thus, the present study aims to evaluate the temporal effects (age, period, and cohort) on total and firearm suicides in Brazil and its major regions from 1980 to 2019.

Results:

1. Line 250: could the increase in suicide rates be due to better record acquisition? Does your method correct for this?

Before the descriptive and inferential analyses, we performed the correction of the death records for the quality of the information and under-enumeration of the deaths. There is no consensus in the literature on the best method to correct information quality (Soares et al., 2016; Rockett et al., 2016; Santos et al., 2014). And so we used the method applied by the Brazilian Ministry of Health, which was also used by Dantas et al (2021). And for the sub-enumeration of deaths we used the correction factors that were presented in the studies by Lima et al., 2014. We believe that with this correction process we managed to minimize the possible period effect promoted by the improvement of the information quality. The process of correcting death records is described in the methodology.

Reference

Soares Filho AM, Cortez-Escalante JJ, França E. Review of deaths correction methods and quality dimensions of the underlying cause for accidents and violence in Brazil. Cien Saude Colet. 2016;21:3803-3818.

Rockett IRH, Hobbs G, Leo D, Stack S, Frost JL, Ducatman AM, Kapusta ND, Walker RL. Suicide and unintentional poisoning mortality trends in the United States, 1987-2006: two unrelated phenomena? BMC Public Health. 2010;10:705

Santos SA, Legay LF, Aguiar FP, Lovisi GM, Abelha L, Oliveira SP. Tentativas e suicídios por intoxicação exógena no Rio de Janeiro, Brasil: análise das informações através do linkage probabilístico. Cad Saude Publica; 2014;30:1057-1066.

Brasil, 2017

Lima EEC, Queiroz BL. Evolution of the deaths registry system in Brazil: associations with changes in the mortality profile, under-registration of death counts, and ill-defined causes of death. Cadernos de saúde pública.2014; 30:1721-1730.

Dantas ESO, Farias YMF, Resende EB, Santos GWS, Silva PG, Meira KC. Estimates of suicide mortality in women residents in northeast brazilian states from 1996 to 2018. Ciencia & saude coletiva.2021; 26:4795-4804

2. Please correct the labels on the tables to English. Some are translated and some

 are not.

Thank you for your attentive and careful reading, the requested changes have been made.

Discussion

1. Please provide a discussion about why the total suicide rate seemed to have increased and yet firearms suicide seems to have decreased in terms of substitution effect or displacement effect, that is if firearms are harder to obtain people switch to other methods leaving overall rates unchanged. That is while firearms restrictions may lower the rate of firearms suicide it doesn’t lower suicide rates overall. (Langmann 2021, Langmann 2020, Gilmour et al 2018).

Thanks to the reviewer for the valuable contribution, we made the requested modification in the discussion.

We make the following modification to the text.

 It is noteworthy that the present descriptive analysis showed a reduction in mortality due to firearm suicide in Brazil and regions prior to implementing the Disarmament Statute, with the most significant decrease in the last five years (2015-2019). These results suggest that the reduction in suicides by firearms is not only correlated with the Disarmament Statute, but other factors may also have contributed to this reality; future studies should be carried out to understand this reality better.

Authors advocate that measures that reduce the availability of lethal means of perpetrating suicide can reduce suicide rates in communities [1,6,48,50]. However, studies have shown that in the absence of methods, such as firearms, individuals can migrate to alternative methods [10,11,12], which may explain the increase in mortality rates from total suicides in the 2000s and younger cohorts, even with a reduction in the risk of death from suicide by firearms, observed in the present study.

Young men are considered more impulsive and use lethal means to commit suicide. Barriers to accessing firearms, this group of people usually resorts to hanging to commit suicide [10,11,12], an easily accessible method of perpetration, which only has effective prevention measures in institutionalized people. In this direction, a study carried out in Brazil from 1980 to 2019 showed an increase in mortality rates due to suicide by hanging of 10.12% comparing the 1990s with the 2000s and an increase of 51.97% in the coefficients of the period from 2000 to 2009 compared to the period from 2010 to 2019. In the same periods, there was a reduction in firearm suicide rates, respectively, 16.92% and 29.63%. Similar results were observed in all Brazilian regions [57]. However, this study only performed a descriptive analysis, so further studies are needed to corroborate the method substitution hypothesis.

2. It appears that in many countries the proportions or methods of suicide have switched to hanging. Could it be that there is an overall preference change to hanging that has nothing to do with the Disarmament Statute? (De Leo et al 2003)

3. It appears to me that suicide by firearms rate decreases start around 1998 (Figure 3) which is 6 years before the start of the Disarmament Statute (2004). Presumably it would also take some time to register firearms etc. When were the majority of firearms registered? It seems that the reduction in suicide by firearm occurred quite a period of time before the implementation of the legislation suggesting another cause may be responsible.

We thank the reviewer for the attentive and careful review, indeed the drop in mortality from firearm suicides in Brazil and regions started in the late 1990s, which was confirmed by the supplementary material we inserted evaluating suicide rates smoothed by moving averages triennials. And so, the reduction of suicides by firearms is not only related to the implementation of the Disarmament Statute, and other factors must have contributed to this reduction. 

We believe that one of the factors may have been the substitution of suicide perpetration methods. Evaluating suicide rates by hanging, smoothed by moving average, we found progressive increase in suicides by hanging in the period studied, especially since especially since the 1990s (S3). These findings should be better evaluated in future studies through interrupted time series to analyze the mortality trend by firearm suicide and the means of perpetration with the highest incidence in Brazil, before and after the implementation of preventive measures for Suicide implemented in Brazil in the 2000's.

We made changes to the discussion so that this information (reduced in the late 1990s) was included in our discussion.

We make the following modification to the text.

Regarding the analysis of period effects (social, political, economic, cultural events, diagnostic and therapeutic advances of a given period and location that impact all age groups) and taking as a reference the five-year period 1995-1999, we verified a distinct profile between total suicides and firearm suicides. There was a reduction in the risk of death (RR<1) from suicide by firearm for all periods and locations, except for the North region, where the risk was not significant for any five-year period. As for the risk of total suicide it was reduced in the South and Midwest regions in all periods, and in the Southeast region, in the first decade of the 2000s. On the other hand, it was increased for Brazil (2015-2019) and for the North (1980-1994 and 2005-2015) and Southeast (1980-1994 and 2015-2019) regions. Furthermore, it produced an upward age-drift trend for the entire period under study for total suicide, excluding the South region, whose trend remained stable. It is noteworthy that the present descriptive analysis showed a reduction in mortality due to firearm suicide in Brazil and regions prior to implementing the Disarmament Statute, with the most significant decrease in the last five years (2015-2019). These results suggest that the reduction in suicides by firearms is not only correlated with the Disarmament Statute, but other factors may also have contributed to this reality; future studies should be carried out to understand this reality better.

S2. Firearm suicide rates among Brazilian men, smoothed by three-year mean averages, according to major Brazilian regions, from 1980 to 2019.

S3 Fig. Suicide by hanging rates among Brazilian men, smoothed by three-year mean averages, according to major Brazilian regions, from 1980 to 2019.

4. If the Disarmament Statute raise the age of acquisition from 21 to 25 should a larger reduction in suicide by firearm appear in that group than other groups as this group was prohibited from acquiring firearms?

Thanks to the reviewer's comment, a greater reduction was actually expected in the 20-24 age groups, however, the mean mortality rates adjusted for the period and cohort effect, show increased rates in the 20-24 and 25-year age groups. 29 years old. We believe that these findings may be related to the fact that the Disarmament statute has reduced access to firearms in the legal market, and that these young people may have accessed this medium through the illegal market. Thus, other policies would be necessary to reduce mortality due to firearm suicide in this age group. 

We make the following modification to the text.

In Brazil, we expected a greater reduction in firearm suicide rates among men aged 20-24 years compared to other age groups, as the Disarmament Statute raised the age for purchasing weapons from 21 to 25 years. However, our findings show higher rates in the 20-24 to 25-29 age groups, suggesting that restricting legal access to firearms was not enough to promote the reduction of suicide in younger age groups. We conjecture that this subpopulation has greater ease of access and purchase in the illegal market, including online – a hypothesis that needs to be investigated in future studies. And so, policies are needed to control gun smuggling in the country, intersectoral policies to prevent teenagers and young adults from getting involved in organized crime, associated with increasing youth participation in sports and cultural activities [10,11,55,56].

5. Line 489: this study cannot demonstrate causation and therefore any claims that the Disarmament Statue was able to reduce suicide by firearms cannot be made. Line 522 last sentence, this claim can also not be made.

We deleted these paragraphs and made changes to the discussion and conclusion which are listed below.

We make the following modification to the text.

Discussion

The main contribution of this study resides in the fact that it evaluated mortality from overall suicide and suicide by firearms under the focus of temporal factors (age, period, and cohort), for Brazil and its major regions. And it showed that the reduction observed in firearm suicide rates from the end of the 1990s and in the cohorts from the 1960s onwards was not enough to reduce mortality from total suicides in the same period and in younger cohorts. These results point to the possibility of replacing suicide methods, a hypothesis that should be studied in future studies, using interrupted time series, or quasi-experimental studies using analysis or differences-in-differences including socioeconomic and demographic variables. The diseases of despair are correlated com unemployment, low income, food insecurity, difficulty in accessing Public Health, Education and Social Assistance Policies And if intersectoral public policies are not implemented to remedy these problems, we believe that suicide rates in Brazil will continue to increase [60,68,69,70].

Conclusion

Reducing access to lethal means of suicide is an important initiative to reduce suicides. However, this protective measure alone is not enough to reduce suicide rates, as methods may be replaced, especially in individuals with psychiatric illnesses who are more impulsive [10,11,12].To prevent suicides in Brazil are needed to public policies increase access to employment and income, psychosocial care, and educational programs in schools to teach children and adolescents to improve their coping strategies to deal with stressful situations [10,11]. In addition to training health professionals, family members, and educators to identify individuals at risk for diseases of despair so that they can be referred for treatment promptly.

Figure 1 etc.

1. Please provide a legend to determine what the different colors represent etc.

We made the changes requested in figures 1 and 3.

Figure 1

Figure 3

---

## [Decision Letter · Decision Letter 1]

6 Mar 2023

PONE-D-22-26688R1The influence of the age-period-cohort effects on male suicide in Brazil from 1980 to 2019PLOS ONE

Dear Authors,

Thank you for submitting your manuscript to PLOS ONE. After careful consideration, we feel that it has merit but does not fully meet PLOS ONE’s publication criteria as it currently stands. Therefore, we invite you to submit a revised version of the manuscript that addresses the points raised during the review process.

There are few weaknesses in the results that must be addressed in full before the article can be reconsidered for publication.

We look forward to receiving your revised manuscript.

Kind regards,

Fernanda Penido Matozinhos, Ph.D

Academic Editor

PLOS ONE

Journal Requirements:

Additional Editor Comments (if provided):

Dear Editor and Author,

Thank you for the opportunity to review this manuscript. I am grateful for the invitation.

After careful consideration, I feel the manuscript explores a very important topic. The questions were responded and modifications in the text made the manuscript come to a satisfying result.

There are few weaknesses in the results that must be addressed in full before the article can be reconsidered for publication.

Kind regards,

Reviewers' comments:

Reviewer's Responses to Questions

**Comments to the Author**

1. If the authors have adequately addressed your comments raised in a previous round of review and you feel that this manuscript is now acceptable for publication, you may indicate that here to bypass the “Comments to the Author” section, enter your conflict of interest statement in the “Confidential to Editor” section, and submit your "Accept" recommendation.

Reviewer #3: All comments have been addressed

2. Is the manuscript technically sound, and do the data support the conclusions?

Reviewer #3: Yes

3. Has the statistical analysis been performed appropriately and rigorously? 

Reviewer #3: Yes

4. Have the authors made all data underlying the findings in their manuscript fully available?

Reviewer #3: Yes

5. Is the manuscript presented in an intelligible fashion and written in standard English?

Reviewer #3: Yes

6. Review Comments to the Author

Reviewer #3: The manuscript presented the effects of age, period, and cohort (APC) on total and firearm-related suicides in men in Brazil and its major regions from 1980 to 2019. It’s a well-written text and detailed document, mainly the methods. I need to highlight the importance of analyzing this information for creating and implementing public policies about health in Brazil. I agree with the changes made by the authors, guided by the reviewers, and I suggest only minor points that need revision.

Minor points

Table 2: correct the caption and put the abbreviations that you use in the table.

I miss a comment at the end of the discussion about the period of the Covid-19 pandemic. You mention the need for future research, so I think it's interesting to add it, mainly because many of the causes of suicide were massive present during the pandemic period.

7. PLOS authors have the option to publish the peer review history of their article (what does this mean?). If published, this will include your full peer review and any attached files.

Reviewer #3: **Yes: **Luana Lara Rocha

---

## [Author Response · Author response to Decision Letter 1]

7 Mar 2023

• Reviewer #3: The manuscript presented the effects of age, period, and cohort (APC) on total and firearm-related suicides in men in Brazil and its major regions from 1980 to 2019. It’s a well-written text and detailed document, mainly the methods. I need to highlight the importance of analyzing this information for creating and implementing public policies about health in Brazil. I agree with the changes made by the authors, guided by the reviewers, and I suggest only minor points that need revision.

Minor points

• Table 2: correct the caption and put the abbreviations that you use in the table.

We carry out the requested request, replace the acronym MRC with TC , which is in the table.

Change in the tables

Dear Editor,

We would like to take this opportunity to thank the anonymous

reviewers that have taken part in this review process for their comments regarding the

manuscript.

All comments were relevant and incorporating the suggested changes has improved the manuscript. The new revised version of the the manuscript has followed the recommendations of Plos One Observation: The sentences in yellow have been removed or replaced by sentences in green. Some sentences were added in blue.

Dear Authors,

Thank you for submitting your manuscript to PLOS ONE. After careful consideration, we feel that it has merit but does not fully meet PLOS ONE’s publication criteria as it currently stands. Therefore, we invite you to submit a revised version of the manuscript that addresses the points raised during the review process.

There are few weaknesses in the results that must be addressed in full before the article can be reconsidered for publication.

We appreciate the valuable contributions, revised the entire results section, and made necessary corrections to the tables (header of some columns and table caption) , as shown below.

Journal Requirements:

Please review your reference list to ensure that it is complete and correct. If you have cited papers that have been retracted, please include the rationale for doing so in the manuscript text or remove these references and replace them with relevant current references. Any changes to the reference list should be mentioned in the rebuttal letter that accompanies your revised manuscript. If you need to cite a retracted article, indicate the article’s retracted status in the References list and also include a citation and full reference for the retraction notice.

All articles used in the manuscript were reviewed, we included the doi or PMI of each of these, and none of the articles have been retracted.

• Reviewer #3: The manuscript presented the effects of age, period, and cohort (APC) on total and firearm-related suicides in men in Brazil and its major regions from 1980 to 2019. It’s a well-written text and detailed document, mainly the methods. I need to highlight the importance of analyzing this information for creating and implementing public policies about health in Brazil. I agree with the changes made by the authors, guided by the reviewers, and I suggest only minor points that need revision.

Minor points

• Table 2: correct the caption and put the abbreviations that you use in the table.

We carry out the requested request, replace the acronym MRC with TC , which is in the table.

• I miss a comment at the end of the discussion about the period of the Covid-19 pandemic. You mention the need for future research, so I think it's interesting to add it, mainly because many of the causes of suicide were massive present during the pandemic period.

We have included the suggested paragraph at the end of the discussion, before presenting the limitations of the study. As described below.

Change in the text

These results point to the possibility of replacing suicide methods. This hypothesis should be studied in future studies, using interrupted time series or quasi-experimental studies using analysis or differences-in-differences, including socioeconomic and demographic variables. Furthermore, studies are needed to assess the impact of the COVID-19 pandemic on suicide mortality in men according to temporal factors (age, period, and cohort) and methods used. An increase in suicide rates is expected due to social isolation, fear of the unknown, fear of contaminating oneself or family members, socioeconomic impacts, such as unemployment, food insecurity, and lack of medical assistance to manage psychosocial problems [70,80 -81].

---

## [Decision Letter · Decision Letter 2]

28 Mar 2023

The influence of the age-period-cohort effects on male suicide in Brazil from 1980 to 2019

PONE-D-22-26688R2

Dear Dr. Karina Cardoso Meira,

We’re pleased to inform you that your manuscript has been judged scientifically suitable for publication and will be formally accepted for publication once it meets all outstanding technical requirements.

Kind regards,

Fernanda Penido Matozinhos, Ph.D

Academic Editor

PLOS ONE

Additional Editor Comments (optional):

Dear authors,

The manuscript explores a very important topic and it has technical rigor.

Thank you for submitting your manuscript to PLOS ONE and making substantial changes in order to improve the manuscript.

We recommend its publication and we would suggest updating the abstract and fixing some of the grammar within it.

Kind regards.

Reviewers' comments:

Reviewer's Responses to Questions

**Comments to the Author**

1. If the authors have adequately addressed your comments raised in a previous round of review and you feel that this manuscript is now acceptable for publication, you may indicate that here to bypass the “Comments to the Author” section, enter your conflict of interest statement in the “Confidential to Editor” section, and submit your "Accept" recommendation.

Reviewer #2: All comments have been addressed

Reviewer #3: All comments have been addressed

2. Is the manuscript technically sound, and do the data support the conclusions?

Reviewer #2: Yes

Reviewer #3: Yes

3. Has the statistical analysis been performed appropriately and rigorously? 

Reviewer #2: Yes

Reviewer #3: Yes

4. Have the authors made all data underlying the findings in their manuscript fully available?

Reviewer #2: Yes

Reviewer #3: Yes

5. Is the manuscript presented in an intelligible fashion and written in standard English?

Reviewer #2: Yes

Reviewer #3: Yes

6. Review Comments to the Author

Reviewer #2: I must apologize for not replying earlier to the authors. For whatever reason, the emails from PLOS ONE were being treated as spam by my server.

Overall the authors have responded in a satisfactory manner. However I would suggest updating the abstract and fixing some of the grammar within it.

Abstract

The Discussion about the disarmament act not associated with a reduction in firearms suicide in ages 20-24 as well as a downward trend prior to the act, and substitution to other methods (paragraphs: 426-468) should be incorporated into the Abstract.

Reviewer #3: (No Response)

7. PLOS authors have the option to publish the peer review history of their article (what does this mean?). If published, this will include your full peer review and any attached files.

Reviewer #2: No

Reviewer #3: **Yes: **Luana Lara Rocha

---

## [Editor Report · Acceptance letter]

4 Apr 2023

PONE-D-22-26688R2 

The influence of the age-period-cohort effects on male suicide in Brazil from 1980 to 2019 

Dear Dr. Meira:

I'm pleased to inform you that your manuscript has been deemed suitable for publication in PLOS ONE. Congratulations! Your manuscript is now with our production department. 

Kind regards, 

on behalf of

Dr. Fernanda Penido Matozinhos 

Academic Editor

PLOS ONE